# Extreme Gradient Boosting Tuned with Metaheuristic Algorithms for Predicting Myeloid NGS Onco-Somatic Variant Pathogenicity

**DOI:** 10.3390/bioengineering10070753

**Published:** 2023-06-23

**Authors:** Eric Pellegrino, Clara Camilla, Norman Abbou, Nathalie Beaufils, Christel Pissier, Jean Gabert, Isabelle Nanni-Metellus, L’Houcine Ouafik

**Affiliations:** 1APHM, CHU Nord, Service d’OncoBiologie, Aix Marseille University, 13015 Marseille, France; clara.camilla@ap-hm.fr (C.C.); nathalie.beaufils@ap-hm.fr (N.B.); christel.pissier@ap-hm.fr (C.P.); isabelle.nanni@ap-hm.fr (I.N.-M.); houcine.ouafik@ap-hm.fr (L.O.); 2CNRS, INP, Inst Neurophysiopathol, Aix Marseille University, 13005 Marseille, France; 3APHM, CHU Nord, Service de Biochimie et de Biologie Moleculaire, Aix Marseille University, 13015 Marseille, France; norman.abbou@ap-hm.fr (N.A.); jean.gabert@ap-hm.fr (J.G.)

**Keywords:** extreme gradient boosting (XGBoost), metaheuristic algorithms, onco-somatic, next-generation sequencing, bioinformatics, myeloid, solid tumors, machine learning

## Abstract

The advent of next-generation sequencing (NGS) technologies has revolutionized the field of bioinformatics and genomics, particularly in the area of onco-somatic genetics. NGS has provided a wealth of information about the genetic changes that underlie cancer and has considerably improved our ability to diagnose and treat cancer. However, the large amount of data generated by NGS makes it difficult to interpret the variants. To address this, machine learning algorithms such as Extreme Gradient Boosting (XGBoost) have become increasingly important tools in the analysis of NGS data. In this paper, we present a machine learning tool that uses XGBoost to predict the pathogenicity of a mutation in the myeloid panel. We optimized the performance of XGBoost using metaheuristic algorithms and compared our predictions with the decisions of biologists and other prediction tools. The myeloid panel is a critical component in the diagnosis and treatment of myeloid neoplasms, and the sequencing of this panel allows for the identification of specific genetic mutations, enabling more accurate diagnoses and tailored treatment plans. We used datasets collected from our myeloid panel NGS analysis to train the XGBoost algorithm. It represents a data collection of 15,977 mutations variants composed of a collection of 13,221 Single Nucleotide Variants (SNVs), 73 Multiple Nucleoid Variants (MNVs), and 2683 insertion deletions (INDELs). The optimal XGBoost hyperparameters were found with Differential Evolution (DE), with an accuracy of 99.35%, precision of 98.70%, specificity of 98.71%, and sensitivity of 1.

## 1. Introduction

Next-generation sequencing (NGS) has revolutionized the field of oncology by enabling high-throughput analysis of cancer genomes. The identification and interpretation of somatic mutations, which are specific to cancer cells and play a crucial role in cancer development, have greatly benefited from NGS technology. In particular, the analysis of myeloid panel mutations has provided valuable insights into myeloid neoplasms, including leukemias and myelodysplastic syndromes, leading to more accurate diagnoses and tailored treatment plans.

To predict the pathogenicity of mutations in the myeloid panel, machine learning models have been employed. In a previous study [1], we developed a random forest model for predicting the pathogenicity of solid tumor mutations. However, due to the differences in the myeloid panel, we have chosen to explore a different machine learning algorithm, Extreme Gradient Boosting (XGBoost), in this work.

XGBoost is a popular gradient boosting algorithm known for its effectiveness and scalability. It has been widely applied in machine learning competitions and real-world applications since its introduction by Tianqi Chen in 2014 [2]. Unlike traditional gradient boosting, XGBoost incorporates L1 and L2 regularization techniques to prevent overfitting, which is a common challenge in gradient boosting models. The L1 regularization encourages the model to select the most important features by adding a penalty proportional to the absolute value of the model’s coefficients. On the other hand, L2 regularization spreads the weight of each feature across all coefficients, reducing the impact of any one feature. Additionally, XGBoost employs shrinkage to improve the model’s generalization ability by slowing down the learning rate.

In this paper, we present a machine learning tool that utilizes XGBoost for predicting the pathogenicity of mutations in the myeloid panel. We aim to optimize the performance of XGBoost by employing metaheuristic algorithms to determine the most suitable architecture and incorporate new data into the learning process. To achieve this, we tuned several hyperparameters, including the number of boosting iterations, learning rate, maximum tree depth, minimum reduction in the loss function, fraction of columns used in each split, minimum number of samples in a leaf node, and fraction of the training set used for each boosting iteration. As the performance of XGBoost is influenced by these hyperparameters, optimizing their selection is crucial. Hence, we employ metaheuristic algorithms, including Differential Evolution (DE), Particle Swarm Optimization (PSO), Genetic Algorithm (GA), and Simulated Annealing (SA), to search the hyperparameter space and identify the combination that maximizes performance.

To evaluate the performance of our model, we conducted 10-fold cross-validation, repeated 3 times, which provides a robust estimate of its predictive power. This rigorous evaluation methodology ensures that the model’s accuracy is accurately assessed on multiple subsets of the data. By combining metaheuristic algorithms, cross-validation, and testing, we establish a powerful framework for optimizing machine learning models and achieving high accuracy in real-world applications.

By leveraging the capabilities of XGBoost and employing metaheuristic optimization, we aim to enhance the prediction of pathogenicity in myeloid panel mutations. The insights gained from this study will contribute to a better understanding of myeloid neoplasms and facilitate personalized treatment strategies based on the genetic profile of individual patients.

## 2. Materials and Methods

The myeloid panel is a critical component in the diagnosis and treatment of myeloid neoplasms, as it allows for the identification of specific genetic mutations that can inform treatment decisions. For example, mutations in the genes FLT3, IDH1, and IDH2 have been associated with poorer outcomes in acute myeloid leukemia (AML) [3,4,5], while mutations in the gene TET2 have been associated with a better response to treatment with the drug azacitidine in patients with myelodysplastic syndromes [6,7]. We use the amplicon sequencing method to target specific genes or regions of the genome that are known to be associated with myeloid neoplasms, such as myeloid leukemias and myelodysplastic syndromes. By sequencing only the relevant regions of the genome, amplicon sequencing can provide a more efficient and cost-effective approach to analyzing the myeloid panel compared with whole-genome sequencing. With this method, a more accurate and comprehensive picture of the genetic mutations present in a patient’s cancer is achieved, which can inform treatment decisions and help monitor disease progression and treatment response over time.

Our dataset is composed of a cohort of 15,977 Single Nucleotide Variants (SNVs), Single Nucleotide Polymorphisms (SNPs), and insertion deletion variants (INDELs) from our 24 genes in the panel, ASXL1, CALR, CBL, CEBPA, CXCR4, DNMT3A, EZH2, FLT3, GATA2, IDH1, IDH2, JAK2, KIT, KRAS, MPL, NPM1, NRAS, PTEN, RUNX1, SF3B1, SRSF2, TET2, TP53, and WT1 note that only the target regions are sequenced), which represents 1050 samples from 2019 to 2022.

For all the clinical samples, the genomic DNA extraction method was performed with the EZ1 XL (Qiagen, Hilden, Germany). We performed sequencing with Ion Torrent S5XL (Thermo Fisher, Waltham, MA, USA) with a sensitivity of 5% and minimum coverage of 300X. Then, sequencing data were analyzed through 2 pipelines. The first pipeline was developed by ThermoFisher on the IonTorrent Suite + Ion Reporter. IonTorrent Suite generates FASTQ data and ensures BAM (binary alignment mapping) alignment with the hg19 reference genome by using the TMAP (Torrent Mapping Alignment Program). The Ion Reporter makes variant caller and variant annotations. The second pipeline was developed in our laboratory and runs open-source software such as BWA-MEM for alignment, SAMtools for mpileup, VarScan2 as the variant caller, and VEP Ensemble for annotations. All data are stored in our local MySQL database. Biologists manually validated variants as pathogenic if the following statements were true:Variant Allele Frequency ≥ 5%;Amino acid change is different from synonymous (≠p.(=)). A synonymous variant will probably have a low influence on the gene because the amino acid does not change;Grantham score is in a range of [5; 215] (in the case of substitution variant): [0, 50] = conservative, [51; 100] moderately conservative, [101; 150] moderately radical, and over 150 radical;Manual inspection on different databases and prediction tools: VarSeak, Varsome, UMD Predictor, Cancer Genome Interpreter (CGI), and OncoKB;We use the tool Integrative Genomics Viewer (IGV) to check if alignment sequences are clear and show no strand bias in the region where the variant is located. It allows us to eliminate false positives;We verify the presence of these pathogenic variants in our second-in-house pipeline to validate them.

With these requirements, biologists determine whether a variant can be validated or rejected as a function of the patient’s pathology.

### 2.1. Programming Language

The programming language used to develop the machine learning model was R version 4.2.2 with XGBoost version 1.6.0.1 and caret version 6.0-93 Libraries for Genetic Algorithms (GA version 3.2.2), Particle Swarm Optimization (pso version 1.0.4), Differential Evolution (DEoptim version 2.2-6), and Generalized Simulated Annealing (GenSA version 1.17).

### 2.2. Data Selection

The myeloid dataset contains clinical information of 15,977 mutation variants (Figure 1) composed of a collection of 13,221 Single Nucleotide Variant (SNVs), 73 Multiple Nucleotide Variant (MNVs), and 2683 insertion deletions (INDELs). All variants were labeled by the biologist as benign or pathogenic in the function of the pathology of the patient. A total of 14,957 variants were labeled as benign (class 0), and 1020 were labeled as pathogenic (class 1). Our dataset is imbalanced. We have a disproportionate representation of the pathogenic. To overcome this problem, there are many appropriate methods to remedy it. We chose to use the oversampling approach for the creation of synthetic data to increase the numbers of the minority class. To perform this operation, we used the package ROSE (random oversampling examples version 0.0-4). In general, oversampling can be a useful technique when working with imbalanced datasets, as it can help to balance the class distribution and improve the performance of machine learning models [8,9]. After applying the oversampling method, our training set was composed of 10,339 occurrences with class 1 (pathogenic) and 10,470 occurrences with class 0 (benign).

However, oversampling can also introduce bias and reduce the overall representativeness of the training data if not carried out carefully. For example, simply replicating the minority class observations can lead to overfitting and poor generalization to unseen data. It is always a good idea to evaluate the performance of a model on a hold-out test set to ensure that it generalizes well to unseen data. This will give a more accurate assessment of the model’s performance on real-world data. For that, we split our dataset into 2 parts, training (70% of the dataset) and testing (the remaining 30%(unseen)). In addition, to avoid overfitting and perform further evaluation and validation, we used 10 cross-validations, repeated 3 times. This ensures that the model generalizes well to new data. Moreover, the results of these evaluations can help to refine the model and improve its performance.

Cross-validation is an important step in the evaluation of a machine learning model. The purpose of cross-validation is to assess how well a model will generalize to an independent data set, in other words, how well the model will perform on unseen data. By using cross-validation, we can avoid overfitting and obtain a better estimate of the model’s performance [10,11,12,13]. In the machine learning model evaluation, we split the data into two parts: a training set and a test set. The model is trained on the training set and then evaluated on the test set. The issue with this approach is that the test set might be small or not representative of the whole dataset, which can lead to poor generalization. Cross-validation solves this issue by dividing the data into multiple folds and then iteratively training the model on different combinations of folds and evaluating it on the remaining fold. This allows us to obtain a better estimate of the model’s performance, since we are using multiple different training and test sets. It also provides a measure of variability in the model’s performance, which is useful for selecting the best model from a set of candidate models.

### 2.3. Problem Formulation

We consider our problem as supervised learning for a classification task. In input, we have real numbers, and in output, we have a category. The model we are looking for is a function in a discrete space.
(1)$$f:R^{14}⇀R^{2},f\left(x_{1},x_{2},\cdots ,x_{14}\right)=y$$
where $y=\left\{Pathogenic,Benign\right\}$

These data come from the laboratory and are produced by the pipeline of the Ion Torrent sequencer. They are present in the form of a .tsv (tabulated separated values) file, where each line corresponds to a mutation in the myeloid panel. All data were labeled by the biologist as 1 for pathogenic and 0 for benign/uncertain significance (identified by the columns isMut). The XGBoost will take in input with 14 features and 2 possible values as output (Equation (Equation 1), as shown in Table 1).

### 2.4. Data Encoding (Feature Construction)

The laboratory has generated these data using the Ion Torrent sequencer’s pipeline, which is available in the form of a .tsv file containing tabulated separated values. Each line in the file represents a mutation present in the myeloid panel, and the biologist assigned a label of 1 to indicate pathogenic and 0 to indicate benign/uncertain significance (identified by the columns isMut). The input to our XGBoost model consists of 14 features, and the output has two possible values (Table 1 and Equation (Equation 1)). Data processing is crucial for creating a clean, informative, and well-structured dataset that can help XGBoost to learn patterns and make accurate predictions. It helps to ensure that the data is of good quality and free of errors, inconsistencies, or missing values.

As in our previous deep learning study to predict the pathogenicity of a BRCA1 and BRCA2 gene mutation, we introduce the information of the amino acid change [14]. A change in the DNA sequence can result in a mutation in which an amino acid is replaced with a different one. Not all amino acid replacements have the same impact on the function and structure of the protein. This depends on the similarity or differences of the replacements, as well as their position in the sequence or structure. The similarity between amino acids can be determined based on substitution matrices, physical and chemical distance, or their basic properties such as size and charge [15]. When an amino acid is replaced by another with similar properties, it is called a conservative replacement, and it rarely leads to dysfunction in the protein. However, when an amino acid is changed to one with different properties, it can result in structural changes in the protein and affect its function, leading to changes in the phenotype and often being pathogenic. For example, a mutation at position 6 that changes a glutamic acid (negative charge) to a valine (uncharged) is known to occur in human sickle cell anemia. Some amino acid modifications at the protein level can be tolerated by the cell without causing any harm, which explains why many “missense” type sequence variations do not have a pathogenic effect and are therefore a significant part of polymorphisms. However, depending on the location of the affected amino acid, missense mutations can have deleterious effects such as altering protein folding, stability, functional domains, or interaction sites with other proteins, leading to either a loss or gain in function. In addition to this information, we grouped all amino acids according to their charge. Each chain represents a chapter in which the amino acids are grouped (Table 3, Figure 2). What we want to see is if the change in polarity of a mutation is taken into account by the machine learning algorithm. Indeed, the chemistry of amino acid side chains is critical to protein structure, because these side chains can bond with one another to hold a length of protein in a certain shape or conformation. Charged amino acid side chains can form ionic bonds, and polar amino acids are capable of forming hydrogen bonds [16,17].

The interactions between amino acid side chains are critical to stabilizing protein structure. They note that polar amino acid side chains can form hydrogen bonds with one another, while charged amino acid side chains can form ionic bonds. They also note that disulfide bonds, which form between two cysteine residues, are another important type of interaction that can stabilize protein structure [17]. One example of a pathogenic mutation involving an amino acid change in a myeloid gene is the NPM1 mutation. NPM1 (nucleophosmin 1) is a gene that is commonly mutated in acute myeloid leukemia (AML). The NPM1 mutation is a missense mutation that results in the substitution of an aspartic acid residue with an alanine residue. This change in the amino acid can lead to changes in the localization and function of NPM1, which contributes to the development of AML. The NPM1 mutation is associated with a favorable prognosis in some subtypes of the disease.

However, it is important to note that the impact of an amino acid change on the function of a protein is not solely determined by the charge of the amino acid but also by other factors, such as the location of the change in the protein and its effect on the protein’s structure and function.

We also decided to introduce information about the Grantham score. It is a measure of the genetic distance between two DNA or protein sequences. It was introduced by Grantham in 1974 as a method for quantifying the differences between amino acids in proteins [18]. The score is calculated based on the physical and chemical differences between amino acids. It takes into account properties such as size, polarity, and charge [18]. Distance difference D for each pair of amino acids i and j are calculated as
(2)$$D_{i,j}=\left[\alpha {\left(c_{i}-c_{j}\right)}^{2}+\beta {\left(p_{i}-p_{j}\right)}^{2}+\gamma {\left(v_{i}-v_{j}\right)}^{2}\right]$$
where *c* = composition, *p* = polarity, and *v* = molecular volume, and they are constants of squares of the inverses of the mean distance for each property. The score is calculated as the sum of the absolute differences between corresponding properties of the two amino acids, normalized by the maximum possible score. The Grantham score ranges from 0 to 215, with higher scores indicating greater differences between amino acids. A score of 0 indicates that the two amino acids are identical, while a score of 215 indicates the maximum possible difference between two amino acids. The Grantham score is often used in bioinformatics to assess the functional similarity between two protein sequences, with lower scores indicating greater similarity. It can also be used to predict the effects of missense mutations, which are mutations that change a single nucleotide in the DNA sequence and result in a different amino acid in the protein. In general, missense mutations with higher Grantham scores are more likely to have a deleterious effect on protein function.

### 2.5. Correlation

The goal of plotting the correlation of variables (also known as a correlation matrix) in the function of the output is to understand the relationship between the predictor variables and the outcome variable. This plot shows the strength and direction of the linear relationship between each pair of variables, as well as the relationship between each variable and the outcome variable. By visualizing the correlation matrix, we can identify which predictors are strongly correlated with the outcome variable, which may suggest that they are important predictors for our model. We can also identify predictors that are highly correlated with each other, which can indicate issues of multi-collinearity, and may suggest the need for variable selection or feature engineering to reduce the number of highly correlated predictors in our model. Additionally, we can use this plot to identify potential interactions between predictors and the outcome variable.

There seems to be a considerable positive correlation between the amino acid mutated chemical value (aamutChemicalVal), the exon, the Coverage, and the amino acid reference chemical value (aarefChemical) with the output variable isMut (Figure 3).

If we want to visualize the distribution of a variable or a set of variables in our dataset to get a sense of the distribution of each variable and how it varies depending on the value of the isMut column (Figure 4), we need to plot the density.

The plot shows the shape of the distribution of data and can reveal patterns that may not be apparent with summary statistics, such as mean and standard deviation. Density plots are especially useful for identifying whether a distribution is symmetric or skewed, whether there are any peaks or gaps in the data, and whether there are any outliers or extreme values. By plotting the density of each variable in function of the output, we can also examine how the distributions of each variable differ for the different output values and potentially gain insight into the relationship between the predictors and the outcome variable, as the density plots are to get a sense of the distribution of each variable and how it varies depending on the value of the isMut column. If there is a clear separation between the two colors for a particular variable, it may suggest that this variable is strongly predictive of the isMut column. Conversely, if the two colors are mostly mixed for a variable, it may suggest that this variable is not strongly predictive. According to the Figure density (Figure 4), we can observe that the variable MAFbin (Minor Allele Frequency) suggests that this variable is strongly predictive. However, for the others variables, it is difficult to suggest that one of them is strongly predictive at this stage.

### 2.6. Principal Component Analysis (PCA)

PCA is a useful tool for dimension reduction and can help identify patterns and relationships among variables in our dataset. It can be a useful way to reduce the dimensionality of our data and identify the most important factors that explain the variation in our dataset [19,20,21,22]. This can help simplify our analysis and make it easier to identify patterns and relationships that are not immediately apparent in the raw data.

In a 2D plot and 3D plot resulting from PCA, we can visually inspect if there is clustering by looking for groups or clusters of data points that are close together. If there are distinct groups of data points that are separated from each other on the plot, this suggests that there may be clustering in the data. As we can observe in Figure 5, the isMut variables (0 and 1), represented by the colors green and red, are mixed and not separated into distinct groups, so it is unlikely that we have clustering.

### 2.7. Metaheuristic Algorithms

Metaheuristic algorithms are a class of optimization algorithms used to solve complex problems where traditional optimization methods are not suitable [23,24,25]. They are called “metaheuristics” because they are higher-level algorithms that are designed to work with a wide range of optimization problems, rather than being tailored to a specific problem.

These algorithms are typically based on simple principles from natural systems, such as genetic evolution or swarm behavior, and use these principles to guide the search for the optimal solution. Unlike traditional optimization methods, metaheuristic algorithms do not rely on explicit mathematical models of the problem and are therefore able to handle problems with high degrees of complexity, uncertainty, and nonlinearity. Metaheuristics can be an efficient way to use trial and error to produce acceptable solutions to a complex problem in a reasonably practical amount of time [26]. In this paper, we confront 4 metaheuristic algorithms (Differential Evolution, Particle Swarm Optimization, Genetic Algorithm, and Simulated Annealing) to tune the hyperparameters of XGBoost to have the optimal solution to predict the pathogenicity of our mutation variants in the myeloid panel.

### 2.8. Defining the Fitness Function

In our specific case, we want to maximize the function accuracy. Our objective function is measuring the performance of XGBoost in terms of accuracy; then, we can represent it mathematically using the following notation:

Let note $D=\left\{(x_{1},y_{1}0,\cdots ,\left(x_{n},y_{n}\right)\right\}$ be the training dataset, where each $x_{i}$ is a feature vector and each $y_{i}$ is the corresponding class label. Let $f(x_{i})$ be the XGBoost model with parameters θ that maps a feature vector *x* to a predicted class label, and let $y^{i}=f\left(x_{i}\right)$ be the predicted class label for the ith example. Then, the accuracy of the XGBoost model with parameters θ can be defined as follows:(3)$$Accuracy\left(\Theta \right)=\left(\frac{1}{n}\right)\times \sum_{i}^{}=1^{n}\times \left[y^{i}=y_{i}\right]$$
where $\left[y^{i}=y_{i}\right]$ is an indicator function that equals 1 if $y^{i}=y_{i}$ and 0 otherwise. The notation $\left[y^{i}=y_{i}\right]$ is sometimes written as a Kronecker delta function $\delta (y^{i},y_{i})$, which is defined as
(4)$$\delta \left(y^{i},y_{i}\right)=\left\{\begin{array}{c} 1\;\mathrm{if}\;y^{i}=y_{i}\\ 0\;\mathrm{otherwise} \end{array}\right.$$

If we reduce Equation (Equation 3) to our binary classification task, then the accuracy of the XGBoost model with parameters θ can be defined as the fraction of correctly classified examples in the training dataset, which can be written as
(5)$$Accuracy\left(\theta \right)=\frac{TP+TN}{TP+TN+FP+FN}$$
where *TP* is the number of true positives (i.e., the number of examples that are correctly classified as positive), *TN* is the number of true negatives (i.e., the number of examples that are correctly classified as negative), *FP* is the number of false positives (i.e., the number of examples that are incorrectly classified as positive), and *FN* is the number of false negatives (i.e., the number of examples that are incorrectly classified as negative). These values can be computed using the confusion matrix, which is a table that summarizes the number of true and false positives and negatives for a classifier
(6)$$\left(\begin{array}{ccc} & \mathrm{Actual}\;\mathrm{Positive} & \mathrm{Actual}\;\mathrm{Negative}\\ \mathrm{Predicted}\;\mathrm{Positive} & TP & FP\\ \mathrm{Predicted}\;\mathrm{Negative} & FN & TN \end{array}\right)$$

Using the confusion matrix (Equation (Equation 6)), we can write
(7)$$\left\{\begin{array}{c} TP=\sum_{i}^{}1^{n}\left[y^{i}=1\;\mathrm{and}\;y_{i}=1\right]\\ TN=\sum_{i}^{}1^{n}\left[y^{i}=0\;\mathrm{and}\;y_{i}=0\right]\\ FP=\sum_{i}^{}1^{n}\left[y^{i}=1\;\mathrm{and}\;y_{i}=0\right]\\ FN=\sum_{i}^{}1^{n}\left[y^{i}=0\;\mathrm{and}\;y_{i}=1\right] \end{array}\right\}$$
where *n* is the number of examples in the training dataset.

Once we compute the values of *TP*, *TN*, *FP*, and *FN*, we can plug them into the accuracy formula to obtain the accuracy (Equation (Equation 5)) of the XGBoost model with parameters θ.

### 2.9. Differential Evolution (DE)

Differential Evolution (DE) is a metaheuristic optimization algorithm that is used to find the global optimum of a given objective function. The algorithm is inspired by the natural process of evolution, where populations of candidate solutions are evolved through generations by mutation, crossover, and selection operations. It was introduced by Storn and Price in 1996 [27]. It was developed to optimize real-parameter and real-valued functions.

The general problem formulation is as follows: For an objective function $f:X\subset R^{D}\to R$ where the feasible region X≠0, the minimization problem is to find $x^{*}\in X$ such that $f\left(x^{*}\right)\leq f\left(x\right)\forall x\in X$ where $f(x^{*})\neq -\infty$.

The optimization problem can be defined as follows: Minimize f(x), where $x=(x_{1},x_{2},\cdots ,x_{n})$ is a vector of decision variables, and f(x) is an objective function that maps the decision variables to a scalar value. The basic idea behind DE is to create a population of candidate solutions, called “agents”, that are randomly generated in the search space. Then, for each generation, the algorithm updates the agents by selecting three different agents, called “parents”, from the current population.

Creating a new agent, called the “trial agent”, by adding a weighted difference between two parents to a third parent.If the trial agent is better than the corresponding parent, then it is selected as the new agent in the population.

The process is repeated for a number of generations until convergence or a stopping criterion is met. Mathematically, the Differential Evolution algorithm can be defined as
(8)$$x_{i}\left(t+1\right)=x_{r1}+F\times \left(x_{r2}-x_{r3}\right)$$
where $x_{i}\left(t+1\right)$ is the new trial agent, $x_{r1},x_{r2}$, and $x_{r3}$ are the three selected parents, and F is a weighting factor that determines the magnitude of the difference between the parents. The performance of Differential Evolution with XGBoost depends on several factors, such as the choice of the objective function, the size of the population, the choice of the weighting factor, and the stopping criteria. In general, DE has been shown to perform well for optimization problems with many local minima, as it can escape from local minima and find the global optimum.

### 2.10. Applying DE to XGBoost

We applied DE to tune the hyperparameters to the XGBoost model for our binary classification. DE is a population-based algorithm that searches for the optimal solution by combining and evolving candidate solutions in the population. To apply DE to the task of tuning XGBoost parameters, we start with an initial population of candidate solutions (i.e., sets of parameter values) and iteratively evolve the population by generating new candidate solutions based on the current population and their fitness (i.e., accuracy). The DE algorithm works as follows:1.Initialize a population of N candidate solutions $\theta_{1}$, $\theta_{2}$, …, $\theta_{N}$, where each $\theta_{i}$ is a set of parameter values.2.Set the crossover rate (CR) and the scaling factor F.3.Repeat the following steps until the stopping criterion is met:(a)For each candidate solution $\theta_{i}$, randomly select three other candidate solutions $\theta_{j}$, $\theta_{k}$, and $\theta_{l}$ from the population, such that j, k, l ≠ i.(b)Generate a new candidate solution $\theta^{\prime }$ by applying the DE mutation operator to $\theta_{j}$, $\theta_{k}$, and $\theta_{l}$. The mutation operator generates a new solution by adding a scaled difference between $\theta_{j}$ and $\theta_{k}$ to $\theta_{l}$, according to the formula $\theta^{\prime }\;=\;\theta_{l}+F\times \left(\theta_{j}-\theta_{k}\right)$, where F is the scaling factor.(c)Apply the DE crossover operator to combine $\theta_{i}$ and $\theta^{\prime }$ into a trial solution $\theta^{\prime \prime }$. The crossover operator generates a new solution by randomly selecting one parameter value from $\theta^{\prime \prime }$ and copying it to the corresponding position in $\theta_{i}$, with a probability of CR.(d)Evaluate the fitness (i.e., accuracy) of $\theta^{\prime \prime }$ and $\theta_{i}$.(e)If the fitness of $\theta^{\prime \prime }$ is better than the fitness of $\theta_{i}$, replace $\theta_{i}$ with $\theta^{\prime \prime }$ in the population.(f)If the fitness of $\theta^{\prime \prime }$ is worse than the fitness of $\theta_{i}$, keep $\theta_{i}$ in the population.(g)If the fitness of $\theta^{\prime \prime }$ is equal to the fitness of $\theta_{i}$, choose one of them at random to keep in the population.(h)Repeat steps (a) to (f) for all candidate solutions in the population.(i)Update the best solution found so far based on the candidate solutions in the population.

In our specific case, we want to maximize the function accuracy. Our objective function is measuring the performance of XGBoost in terms of accuracy, then we can represent it mathematically using the Equation (Equation 3).

Once we compute the values of TP, TN, FP, and FN, we can plug them into the accuracy formula to obtain the accuracy(Equation (Equation 5)) of the XGBoost model with parameters θ. The objective of the DE algorithm is to find the values of θ that maximize the accuracy function (Equation (Equation 5)). This is achieved by iteratively evolving the population of candidate solutions using the mutation and crossover operators and selecting the best solution found so far.

We fixed the parameters as the following:Mutation factor = 0.5.Crossover probability = 0.9.Strategy = 1.Population size = 10.Number of iterations = 10.

### 2.11. Genetic Algorithm (GA)

The GA was introduced by J. Holland [28], and it is based on the natural evolution theory of Darwin. It is a metaheuristic algorithm that can be used to solve complex optimization problems by mimicking the process of natural selection and genetic recombination. The basic idea behind GA is to maintain a population of candidate solutions to a problem and iteratively improve the quality of the solutions in the population through selection, crossover, and mutation operations [11,28,29,30,31]. We start by initializing a population of m such binary strings randomly, where each string represents a possible solution to the problem. We evaluate each string in the population using an objective function, which gives a score or fitness value indicating how well the string solves the problem.

The Genetic Algorithm then proceeds through a series of iterations or generations. In each generation, we use selection to choose a subset of strings from the current population to serve as parents for the next generation. Selection is typically conducted probabilistically based on the fitness values of the strings, with fitter strings being more likely to be selected. We use the crossover to create new strings by combining genetic material from pairs of parents. Crossover involves selecting a crossover point in the strings and swapping the bits before and after that point to create two offspring strings. Finally, we use mutation to introduce small random changes into the offspring strings, to add diversity to the population, and prevent premature convergence to suboptimal solutions.

After each generation, we evaluate the fitness of the new population and repeat the process until some stopping criterion is met (e.g., a maximum number of iterations or a satisfactory level of fitness is achieved). The goal is that the population evolves towards better and better solutions over time through the combination of selection, crossover, and mutation.

### 2.12. Applying GA to XGBoost

To use GA to tune the XGBoost parameters, we first need to define the parameter space and the objective function to be optimized. The parameter space defines the range of values that each parameter can take, and the objective function measures the performance of the XGBoost algorithm for a given set of parameters. The objective function is defined in terms of accuracy metrics (Equation (Equation 5)). Next, we create a population of candidate parameter sets and evaluate their performance using the objective function. The selection, crossover, and mutation operations are then used to generate a new population of candidate parameter sets, which are evaluated again. The process continues until a satisfactory set of parameters is found or a predefined stopping criterion is met.

In our case, the chromosome structure would be a string of 7 genes (eta, nrounds, max_depth, gamma, colsample_bytree, min_child_weight, and subsample), where each gene represents a hyperparameter value. For example, we have the following hyperparameter values:1.$eta=[\alpha_{1},\alpha_{2},\alpha_{3}]$.2.$nrounds=[\beta_{1},\beta_{2},\beta_{3}]$.3.$max\_depth=[\delta_{1},\delta_{2},\delta_{3}]$.4.$gamma=[\gamma_{1},\gamma_{2},\gamma_{3}]$.5.$colsample\_bytree=[\lambda_{1},\lambda_{2},\lambda_{3}]$.6.$min\_child\_weight=[\psi_{1},\psi_{2},\psi_{3}]$.7.$subsample=[\phi_{1},\phi_{2},\phi_{3}]$.

Then, a possible chromosome structure could be a string of 7 genes, where each gene represents an index of the corresponding hyperparameter value. For example, a chromosome could look like this [1,2,3,1,2,1,3], where the first gene (1) represents the eta value of $\alpha_{1}$, the second gene (2) represents the nrounds value of $\beta_{2}$, and so on. This way, each chromosome represents a unique combination of hyperparameter values, and the GA algorithm can search for the optimal combination by evolving the population of chromosomes over multiple generations. The fitness function (Equation (Equation 3)) takes a chromosome and uses the corresponding hyperparameter values to train and evaluate an XGBoost model on a validation set. The fitness score returned by the fitness function would reflect the performance of the model, and this score would be used to evaluate the fitness of the individual (i.e., chromosome) in the population. The individuals with the highest fitness scores would be selected for reproduction and used to generate new offspring (i.e., new candidate solutions) for the next generation.

The Parameters used for the Genetic Algorithm can be found in Table 4.

### 2.13. Particle Swarm Optimization (PSO)

Particle Swarm Optimization (PSO) is one of many optimization algorithms that are inspired by nature. PSO simulates the behavior of a flock of birds or a school of fish. It shares many similarities with evolutionary computation techniques, such as Genetic Algorithms (GA) [32,33]. In PSO, we define a swarm of particles that represent potential solutions to the optimization problem. Each particle has a position and a velocity, which are updated based on its previous position, the best position found by any particle in the swarm so far, and the global best position found by the swarm. At the beginning of the algorithm, every particle is positioned in the search space for the problem, either randomly or not. As the algorithm progresses through each iteration, the particles are moved according to three components: their current speed $V_{i}^{t}$, their best solution $pbest_{i}$, and the best solution obtained in their neighborhood gbest. The algorithm will simulate the movement of a swarm to find the best value. For the particles to move, the algorithm uses the following parameters [32]:(9)$$v_{i}^{t+1}=\omega V_{i}^{t}+c_{1}\times r\times \left(pbest_{i}-X_{i}^{t}\right)+c_{2}\times r\times \left(gbest-X_{i}^{t}\right)$$

Each step *t*, the position of the particle *i*, $X_{i}^{t}$ is updated in function of the particle’s velocity $v_{i}^{t}$, and the new position of the particle would be
(10)$$X_{i}^{t+1}=X_{i}^{t}+v_{i}^{t+1}$$

ω is the particle inertia coefficient. This parameter helps the particle to move by inertia toward better positions. *r* is a random value between 0 and 1. $pbest_{i}$ is the best value of the particle and $gesbt_{i}$ is the best value of the swarm. $c_{1}$ is the personal acceleration coefficient and $c_{2}$ is the global acceleration coefficient; they represent the weights of approaching the $pbest_{i}$ and the gbest of the particle. In a significant number of iterations of Equations (Equation 9) and (Equation 10), the particles should converge towards a value that should be the best solution achieved by the swarm.

### 2.14. PSO Applied to XGBoost

To tune the XGBoost algorithm parameters using PSO, we can use a similar approach to the one I described for Genetic Algorithms. In the case of PSO, the chromosome will be the particle. Each particle represents a candidate solution that contains values for each hyperparameter. The particle’s position in the search space corresponds to a set of hyperparameter values, and its velocity determines how it moves through the search space. The objective function (Equation (Equation 3)) is evaluated for each particle’s position, and the particle’s position is updated based on its own best position and the best position of its neighbors. This process is repeated until convergence, hopefully finding a good set of hyperparameter values that optimize the objective function.

To apply PSO to the task of tuning XGBoost algorithm parameters, we would use the accuracy function as the objective function to be maximized. We can represent each particle as a set of parameter values θ=(eta,nrounds,max_depth,gamma,colsample_bytree,min_child_weight,subsample), where each parameter has an associated range of possible values. The PSO algorithm works as follows:1.Initialize the swarm of particles with random positions and velocities.2.Evaluate the accuracy of each particle by training an XGBoost model with the corresponding parameter values and computing the accuracy on the training dataset.3.Update the best position found by each particle and the global best position found by the swarm so far.4.Update the velocity and position of each particle based on its own previous position, the best position found by any particle in the swarm so far, and the global best position found by the swarm.5.Repeat steps 2 to 4 until a stopping criterion is met (the maximum number of iterations is reached).

The accuracy function that we want to maximize using PSO was defined by the Equation (Equation 3), where *D* is our training dataset, where each $x_{i}$ is a feature vector, and each $y_{i}$ is a binary class label (0 or 1). We can also express the accuracy as a function of the confusion matrix C, which is a 2 × 2 matrix that contains the number of true positives, true negatives, false positives, and false negatives:(11)$$Accuracy\left(\theta \right)=\Gamma \left(\theta \right)=\frac{\left(C[1,1]+C[2,2]\right)}{\left(C[1,1]+C[1,2]+C[2,1]+C[2,2]\right)}$$

The goal of the PSO algorithm is to find the set of XGBoost parameters θ that maximize this accuracy function. We fixed the parameters learning rate to 0.5; the parameters phi1 and phi2, known as the cognitive and social parameters, were fixed to 2.05, respectively, and the initial position to random.

### 2.15. Simulated Annealing (SA)

Introduced in 1983, Simulated Annealing (SA) has become a popular tool for tackling both discrete and continuous problems across a broad range of application areas [34,35]. It is a metaheuristic optimization algorithm that is used to find the global minimum of a function [36]. It is inspired by the process of annealing in metallurgy, where a metal is heated and then slowly cooled to reduce its defects and increase its strength. In the Simulated Annealing algorithm, the optimization process is modeled as a “cooling” process, where the objective function is gradually optimized by iteratively adjusting the parameters of the solution.

The algorithm works by starting with an initial solution and iteratively perturbing the solution by randomly adjusting its parameters. If the new solution is better than the previous one, it is accepted as the new current solution. However, if the new solution is worse, it is accepted with a certain probability that decreases as the optimization process continues. This allows the algorithm to explore a wider search space and avoid getting stuck in local minima. The acceptance probability of a worse solution is determined by a temperature parameter, which is gradually reduced over time to decrease the likelihood of accepting worse solutions. As the temperature decreases, the algorithm becomes more and more selective, eventually converging to the global minimum of the objective function. Simulated Annealing has been successfully applied to a wide range of optimization problems, including the traveling salesman problem [37,38], the knapsack problem [34], and various engineering and scientific applications.

### 2.16. Applying SA to XGBoost

In the context of tuning the parameters of an XGBoost model, Simulated Annealing is used to explore the space of possible parameter values and find the optimal combination that maximizes the accuracy of the model on a given dataset *D*. The objective function, in this case, would be the accuracy of the XGBoost model defined by the Equation (Equation 5), and the parameters to be optimized would be eta, nrounds, max_depth, gamma, colsample_bytree, min_child_weight, and subsample. The Simulated Annealing algorithm would explore different combinations of parameter values by iteratively adjusting them and accepting or rejecting the new solution based on the acceptance probability.

To apply SA to the task of tuning XGBoost parameters, we can use a similar approach to the one we described for DE, GA, and PSO. We start with an initial set of parameter values and iteratively explore the parameter space by perturbing the current set of parameter values and accepting or rejecting the new set of values based on a probabilistic criterion. The SA algorithm works as follows:1.Initialize the current set of parameter values θ.2.Set an initial temperature T and a cooling schedule that reduces the temperature over time.3.Repeat the following steps until the stopping criterion is met:(a)Generate a new set of parameter values $\theta^{\prime }$ by perturbing the current set of values.(b)Compute the change in accuracy $\Delta =Accuracy\left(\theta^{\prime }\right)-Accuracy\left(\theta \right)$.(c)If Δ>0, accept the new set of parameter values (i.e., set $\theta =\theta^{\prime }$).(d)If Δ≤0, accept the new set of parameter values with a probability determined by the temperature and the magnitude of Δ.(e)Update the temperature according to the cooling schedule.

The accuracy function that we want to maximize using SA is the same as the one we defined earlier for DE, GA, and PSO (Equation (Equation 3)). The perturbation function Ω that generates a new set of parameter values can be defined based on the current set of parameter values and their ranges. We generate a new set of values by adding a small random perturbation to each parameter value. This function is generated as the following: Let *x* be a vector of hyperparameters, and let δ be a small constant. We can define the perturbation function as
(12)f(x)=x+δ×r
where r is a vector of randomly generated values from a uniform distribution on the interval [−1, 1]. In other words, for each element i in the vector x, we generate a random value $r_{i}$ from the interval [−1, 1] and multiply it by the constant δ to obtain the perturbation value. Then, we add the perturbation value to the original value $x_{i}$ to obtain the new value:(13)$$x_{i^{\prime }}=x_{i}+delta\times r_{i}$$

This perturbation function generates a new set of hyperparameter values by adding a small random perturbation to each hyperparameter value in x. The magnitude of the perturbation is controlled by the constant delta, which can be adjusted to balance the exploration of the search space with the exploitation of promising regions.

## 3. Results

### 3.1. Model Performance

We evaluated the predictive models on validation sets and we compared the results of validation to obtain the best XGBoost architecture for identifying pathogenicity in myeloid panel genes. Finally, we tested the best model based on a test dataset on a new NGS run comprising 6 different patients (total of 89 mutations) to obtain general predictions on onco-somatic mutation variants. To assess the performance of each model, accuracy, recall (sensitivity), precision, sensibility, kappa, error rate, and Matthew Correlation Coefficient (MCC) were calculated to measure the performance of classification models obtained with Differential Evolution, Particle Swarm Optimization, Genetic Algorithm, and Simulated Annealing. These parameters are defined as follows:$Accuracy=\frac{TP+TN}{TP+TN+FP+FN}$.$Precision=\frac{TP}{TP+FP}$.$Recall=\frac{TP}{TP+FN}$.

True positives (*TP*) and true negatives (*TN*) are defined as the number of mutations that are classified correctly as pathogenic and benign. False positives *(FP)* and false negatives (*FN*) are defined as the number of mutations that are misclassified. *FP* is a result where the model incorrectly predicts the positive class. An *FN* is a result where the model incorrectly predicts the negative class. Precision is defined as the number of positive samples the model predicts correctly (true positives) divided by the true positives plus the false positives. It attempts to determine the proportion of positive identifications that were correctly identified by the model. Recall attempts to determine the proportion of actual positives identified correctly. It is defined as true positives divided by true positives plus false positives. Model performance was evaluated using the receiver operating characteristic area under the curve. The receiver operating curve (ROC) is a graph where sensitivity is plotted as a function of 1−specificity. The area under the ROC is denoted AUC. The true positive rate (TPR), also known as sensitivity (or recall), is defined as the percentage of pathogenic mutations that are correctly identified. The specificity or true negative rate (*TNR*) is defined as the percentage of mutations that are correctly identified as benign.



$Specificity=TNR=\frac{TN}{TN+FP}$



The error rate, also known as the misclassification rate, is a commonly used performance metric for binary classification models. It represents the proportion of misclassified samples in the test set, i.e., the fraction of test samples that the model predicted incorrectly. A lower error rate indicates that the model is better at accurately predicting the class of new, unseen data. Calculating the error rate on the test set is important because it provides an unbiased estimate of the model’s generalization performance, i.e., how well the model will perform on new, unseen data. If the model is overfitting to the training set, it may perform well on the training set but poorly on the test set. Calculating the error rate on the test set allows us to detect such overfitting and make necessary adjustments to the model.
$\mathrm{Error}\;\mathrm{Rate}=\frac{\sum_{i=1}^{n}\left[y_{i}\neq \hat{y_{i}}\right]}{n}$
where *n* is the total of predictions, $y_{i}$ is the actual target label for the *i*th observation, and $\hat{y_{i}}$ is the predicted label for the *i*th observation.

Even if we oversampled our dataset during training, it is important to calculate the Matthew Correlation coefficient (MCC). When we oversample our dataset, we are essentially creating synthetic samples in the minority class to balance the distribution of the classes. However, this does not change the performance of the model on the test set, which is evaluated on the original, unmodified distribution of the classes. Therefore, it is still important to calculate the MCC on the test set to assess the true performance of the model on the original, imbalanced distribution. MCC is a measure of the quality of binary (two-class) classifications, which takes into account true and false positives and negatives. It gives more weight to the correctly classified samples in the smaller class.



$MCC=\frac{TP\times TN-FP\times FN}{\sqrt{\left(TP+FP)(TP+FN)(TN+FN)(TN+FP\right)}}$



The F1-score evaluates the ability of a classification model to efficiently predict positive individuals by making a trade-off between precision and recall. It summarizes the precision and recall values into a single metric as the harmonic mean of precision and recall, which translates into the following equation:$$F1-\mathrm{score}=\frac{2}{\frac{1}{\mathrm{precision}}+\frac{1}{\mathrm{recall}}}$$

The equation can be simplified and expressed directly from the components of the confusion matrix:$$F1-\mathrm{score}=\frac{TP}{TP+\frac{1}{2}\left(FN+FP\right)}$$

Finally, we evaluate the kappa metric for each model. The kappa coefficient is frequently used in statistics to test inter-rater reliability [39]. It is a statistical measure used to assess the agreement between two sets of ratings or classifications. It takes into account the amount of agreement that is expected to occur by chance and compares this to the actual level of agreement observed. It is calculated as the ratio of the observed agreement between the two sets of ratings and the expected agreement that would be expected to occur by chance. The expected agreement is calculated based on the marginal totals of the ratings, while the observed agreement is calculated based on the actual ratings. The ranges are from −1 to 1, with a value of 1 indicating perfect agreement, 0 indicating no agreement beyond chance, and negative values indicating less agreement than would be expected by chance (Table 5).

### 3.2. Performance on the Training Set

Summary results with different methods with a 10 cross-validation repeated 3 times. Each algorithm was evaluated 30 times. The training set represents 70% of the dataset, and the remaining 30% is the test set. The seven important hyperparameters to control XGBoost behavior that we tuned are the following:nrounds: The number of boosting iterations to perform. A boosting iteration trains a new weak learner to improve the overall performance of the model.eta: The learning rate or step size for shrinking the contribution of each tree in the model. A smaller eta value will result in more rounds of boosting required to achieve the same level of accuracy, but each boosting iteration will have a smaller impact on the final model.max_depth: The maximum depth of each tree in the model. Deeper trees capture more complex relationships in the data but can also lead to overfitting if they are too deep.gamma: The minimum reduction in the loss function required to split a node in the tree. This parameter controls the trade-off between the complexity of the model and overfitting. A smaller value allows the model to split nodes more frequently, increasing the model’s complexity and potential for overfitting.colsample_bytree: The fraction of columns (features) to be used in each split. This parameter can be used to prevent overfitting by reducing the impact of noisy features.min_child_weight: The minimum number of samples required in a leaf node. This parameter can be used to control overfitting by increasing the minimum number of samples required in each leaf node.subsample: The fraction of the training set to use for each boosting iteration. A smaller value can make the model more robust to noise and prevent overfitting but may also result in a slower training time.

For each metaheuristic algorithm, the range of the XGBoost hyperparameters is defined as follows:1.nrounds = [100, 600].2.eta = [−5, −3].3.max depth = [2, 6].4.gamma (γ) = [0, 1].5.colsample by tree = [0.4, 1].6.min child weight = [1, 3].7.subsample = [0.5, 1].

The results returned by the different metaheuristic algorithms are described in Table 6 and Figure 6, with a population size of 10 and iteration of 10. Only for Simulated Annealing, we fixed the iteration to 100. On a second test, for DE and PSO, we increased the number *n* of the population to 40 to see if we could obtain better precision. Indeed, the choice of population size is one of the factors that can affect the performance of the optimization algorithms. A smaller population size generally results in faster optimization but may not converge to the global optimum solution, while a larger population size may converge to a better solution but may take longer to run. The choice of population size may depend on the problem itself and the characteristics of the search space. For some problems, a larger population size may be more beneficial, while for others, a smaller population size may be sufficient.

In our case, the difference in accuracy between population size (n=10) and population size (n=40) suggests that increasing the population size leads to a more thorough search of the search space, but it also increases the computational time. The increase in population size did not improve accuracy (Figure 7 and Figure 8).

The results show that PSO finds its search space faster than DE. We have a computation time of 6.64 h with a population size of 10 with DE and only 10.21 min with PSO.

We did not test increasing the population size for the GA algorithm, because for a population size of 10, the algorithm is very resource-intensive, and GA is clearly below the performance of DE and PSO for our specific problem. The accuracy after 10 generations was 93.33% (Figure 9 and Table 6), which is a difference of −5.89% with DE and −5.71% with PSO and SA.

There was also no difference in performance between PSO and SA, with an accuracy of 99.04%. The only difference is that SA took more time than PSO (5.30 h against 10.21 min for PSO). Unlike the three other metaheuristic algorithms, we have a linear function when using Simulated Annealing (SA), as it is a stochastic optimization algorithm that gradually reduces the “temperature” parameter over iterations (Figure 10). This temperature reduction allows the algorithm to explore both high- and low-quality solutions and ultimately converge to the global optimum. The linear function in our plot represents the gradual improvement in accuracy as the algorithm progresses through iterations, which is the expected behavior for SA. In the results, DE still has the best result in terms of accuracy in training. We still have to see the performance of these different metaheuristic algorithms on the test set.

The difference in runtime between the metaheuristic algorithms is due to their inherent algorithmic differences. Each algorithm has its way of exploring and exploiting the search space, which affects how long it takes to find a good solution. For example, the Differential Evolution (DE) algorithm uses a population-based approach to search the solution space, which requires generating and evaluating a large number of candidate solutions. This can be time-consuming, especially if the evaluation of each solution is computationally expensive.

On the other hand, the Particle Swarm Optimization (PSO) algorithm uses a simpler approach of iteratively adjusting the positions and velocities of particles in the search space. This approach requires fewer evaluations of candidate solutions and may converge faster, leading to a shorter runtime.

The Genetic Algorithm (GA) also uses a population-based approach similar to DE, but it requires the additional steps of selection, crossover, and mutation. These steps add extra complexity to the algorithm, which may increase its runtime. Finally, the Simulated Annealing (SA) algorithm uses a probabilistic approach to randomly accept or reject new candidate solutions based on a temperature parameter. This approach may require more iterations to converge to a good solution, leading to a longer runtime.

### 3.3. Performance on the Test Set

The performance for the four metaheuristic algorithms on the test set is summarized in Table 7.

#### 3.3.1. Performance of DE on the Test Set

With the tuned hyperparameters found with DE, we had an accuracy of 99.35%, a precision of 98.70%, a specificity of 98.71%, a sensitivity of 100%, and an MCC of 98.70% on the test set. This model is capable of distinguishing between pathogenic and benign mutations. When it predicts that a variant is pathogenic, it was correct 98.70% of the time. An MCC of 98.70% indicates that the model is providing a good balance between true positive and false positive rates.

If we look at the confusion matrix obtained with the DE algorithm, it was capable of correctly classifying 8860 occurrences, and 58 were badly classified by the algorithm. This would not have an impact on clinical outcome, as the biologist manually inspected all variants, and the proportion of a 0.65% error rate is very small in comparison with the efficiency of the classifier in predicting the correct class.
$$\left(\begin{array}{ccc} & Benign & Pathogenic\\ Benign & 4429 & 0\\ Pathogenic & 58 & 4431 \end{array}\right)$$

The area under the ROC curve (AUC) score is significantly good, with a value of 99.78% (Figure 11). This indicates that the model is almost perfect in its ability to separate positive and negative samples. This is a very high score and suggests that the model is performing very well on the task it was trained for.

#### 3.3.2. Performance of PSO on the Test Set

The performance of XGBoost with the hyperparameters found with the PSO is less good in comparison with other classifiers. The error rate reaches a value of 0.96%. The precision remains acceptable with a value of 98.10%. This is still above the classifier obtained with DE and SA but above GA.
$$\left(\begin{array}{ccc} & Benign & Pathogenic\\ Benign & 4402 & 1\\ Pathogenic & 85 & 4430 \end{array}\right)$$

In total, 1 mutation was considered pathogenic when it was benign, and 85 mutations were classified as benign when they were pathogenic. An AUC score of 99.79% and an F1-score of 99.03% suggest that our binary classifier is performing very well on the test set, with high accuracy, precision, and recall. However, these performances remain below those obtained with the hyperparameters found by DE.

The ROC curve is shown in Figure 12.

#### 3.3.3. Performance of GA on the Test Set

Out of the four metaheuristic algorithms, GA is the least successful in prediction performance. With an error rate of 7.49% in a medical diagnosis application, it can be considered relatively high, as it means that almost 1 in 13 patients are misdiagnosed. This could have serious consequences.
$$\left(\begin{array}{ccc} & Benign & Pathogenic\\ Benign & 4039 & 220\\ Pathogenic & 448 & 4211 \end{array}\right)$$

An accuracy of 92.51% and an AUC of 97.90% (Figure 13) suggest that the model is performing reasonably well overall. However, a precision of 90.01% and specificity of 90.02% indicate that there is still room for improvement in reducing false positives and false negatives.

#### 3.3.4. Performance of SA on the Test Set

Our XGBoost model tuned with Simulated Annealing performed quite well on the test set, with high accuracy, precision, specificity, sensitivity, and AUC (Figure 14) and a low error rate (Table 7). The F1-Score is also high, which indicates that the model has a good balance between precision and recall.
$$\left(\begin{array}{ccc} & Benign & Pathogenic\\ Benign & 4410 & 1\\ Pathogenic & 77 & 4430 \end{array}\right)$$

If we look at the confusion matrix, 77 pathogenic occurrences were badly classified, and only 1 benign variant was classified as pathogenic. Comparing the performance with DE, it appears that the XGBoost model tuned with DE has a slightly higher accuracy, precision, specificity, and AUC than the model tuned with Simulated Annealing. However, the model tuned with Simulated Annealing has a slightly higher sensitivity, and the difference in performance between the two models may not be significant.

### 3.4. Feature Importance

The feature importance plot provides information about the contribution of each feature to the prediction of the target variable (the class of the mutation variant). A high score indicates that the feature has a strong impact on the prediction, while a low score indicates that the feature has a weaker impact. The method used to calculate feature importance for XGBoost models is the “weight” method, which simply counts the number of times each feature is used to split the data across all trees in the model. The importance scores are then normalized to sum to one. The method is included in the package Caret.

The contribution of each feature to the model’s prediction is based on the split gain. The split gain is a measure of how much each feature splits the data and leads to a reduction in impurity or increase in accuracy.

To compare the feature importance across the different models obtained with the metaheuristic algorithms, we can visually compare the relative importance of each feature in each model using a bar chart (Figure 15).

The observations based on the feature importance results:

MAFbin is consistently the most important feature across all four models. The top five important features in each model are somewhat similar, but their order of importance varies. The importance of Coverage, Variant Allele Frequency (Freq), and amino acid reference chemical properties (aarefChemical) is consistently high across all models, although their rank order varies. The importance of exons, chromosome number (chr), amino acid before the mutation (aarefbin), and VarEffectBin varies across the models, with some models giving these features higher importance than others. The least important features across all models are ProtBin and TypeBin, although they still contribute to the overall predictive power of the models. These results suggest that some features are consistently more important than others in predicting the outcome variable, but the exact rank order of importance can vary across different models obtained with different metaheuristic algorithms.

### 3.5. Performance on a New Dataset

To enhance the output of the XGBoost model found with the DE algorithm, we decided to use the probability score of a variant mutation class to classify the variant by function of its degree of being pathogenic (Table 8).

We compared our optimized XGBoost algorithm with the Differential Optimization algorithm on a run composed of 6 different samples with a total of 89 mutations. We compared the XGBoost result with the biologist’s decision and three other online prediction tools. An overview of the XGBoost result on the run is available in Table 9. The complete result is available in Appendix A.

Of the 89 mutations, XGBoost agreed 88 times with the biologist’s decision. The point of disagreement between the algorithm and the biologist concerns the mutation on exon 9 of the CALR gene (c.1154A>C/p.Lys385Thr), which is given as pathogenic by the algorithm XGBoost and considered rather as uncertain significance by the biologist. Moreover, there is also a point of discrepancy between the online predictive algorithms on this mutation. For example, the UMD predictor gives it as probably pathogenic, while the Varsome and Cancer Genome Interpreter gives it as a passenger. The biologist’s decision to label it as of uncertain significance corroborates with the decision with sites such as oncokb and CancerVar that classify this mutation as Tier III Uncertain/Unknown Oncogenic Effect. To understand the decision of the XGBoost algorithm to classify the mutation as pathogenic, if we focus on the change in amino acid charge, we have a Lysine which belongs to the group of positively charged polar chains, which gives us a Threonine which belongs to the group of neutral polar chains. A change in amino acid can have a significant impact on the protein’s function and stability. The change from Lysine to Threonine in the CALR gene is a substitution of a positively charged polar amino acid with a neutral polar one. This change could potentially alter the charge distribution and hydrophobicity of the protein and potentially impact its interaction with other proteins or its ability to fold properly. The effect of this change on the protein can be predicted to some extent by bioinformatics tools. However, the ultimate impact of this change may be best determined through experimental validation, such as functional assays or structural studies. Additionally, the specific location of this change, on exon 9 of chromosome 19, may also play a role in its impact. Further studies would be needed to determine the precise effect of this amino acid change.

In general, the impact of an amino acid change on protein function can be assessed through a variety of computational and experimental methods, such as structural modeling, biochemical assays, and functional studies. An important point to highlight is the performance of the XGBoost algorithm to correctly classify the JAK2 (c.1849G>T/p.Val617Phe) hotspot as pathogenic, where predictive algorithms give it as benign. This somatic JAK2 mutation V617F is known to have been reported in most patients with polycythemia vera (PV), as well as in approximately one-third of patients with essential thrombocythemia and or idiopathic myelofibrosis [40].

### 3.6. Test XGBoost Model Tuned with DE on Solid Tumor Hotspots

To test the XGBoost tuned model, we assessed it against 11 mutations known to be hotspots in solid tumors. These mutations are completely outside the training, so it is interesting to identify how the algorithm will behave. On 11 mutations, the algorithm misclassified 2 mutations on the EGFR: p.Thr725Met and p.Thr790Met (Table 10). The algorithm classified 9 out of 11 solid tumor hotspots mutations correctly; then, its success rate on this particular set of mutations would be around 81.8%. We introduced, in Table 10, in addition to XGBoost, the column MLRF, which refers to our random forest algorithm [1] that we developed in the laboratory within the framework of the prediction of solid tumors.

While this success rate may seem high, it is important to keep in mind that the algorithm was not specifically trained on these mutations and that the sample size of 11 mutations is relatively small. We need to evaluate the algorithm’s performance on a larger and more diverse set of mutations to get a better sense of its overall success rate. However, this success rate is quite high from the perspective of training XGBoost on a wider range of mutations.

If we investigate why the model misclassified the two mutations on EGFR: p.Thr725Met and p.Thr790Met, we must refer to the importance of the DE model feature. In the XGBoost model tuned with DE, we know that the feature “MAFbin” has the highest important score of 100.0000, indicating that it was the most important feature for the model to predict the outcomes. “Coverage” comes next, with a score of 42.0798, followed by “aarefChemical” with a score of 27.2689, “Freq” with a score of 22.6953, and “POS” with a score of 14.5312. Other features, such as “Exon”, “aamutbin”, “aarefbin”, “Grantham”, and “aamutChemicalVal”, also have some importance scores, which means that they contributed to the model’s predictions. The difference between the reference amino acid and the mutated amino acid could be another factor why the algorithm misclassified these two EGFR mutations. We know that the EGFR mutation Thr790Met is considered pathogenic because it confers resistance to tyrosine kinase inhibitors (Gefitinib TKIs), which are commonly used to treat non-small-cell lung cancer (NSCLC) patients with activating EGFR mutations. The Thr790Met mutation causes a change in the EGFR protein structure that makes it less sensitive to TKIs. The mutation results in the creation of an additional binding pocket in the EGFR protein that allows it to bind to ATP more tightly, making it less susceptible to inhibition by Gefitinib TKIs. Therefore, while the Thr790Met mutation itself is not directly oncogenic, it is considered pathogenic because it allows tumor cells to continue growing and spreading even in the presence of TKIs, which are the standard of care for EGFR-mutant NSCLC. In some cases, a Thr to Met substitution may be benign or have a minimal impact on the protein’s structure and function [41,42,43]. However, in other cases, it can have significant functional consequences, such as altering the protein’s enzymatic activity, interaction with ligands or other proteins, or cellular localization. Overall, the impact of a Thr to Met substitution depends on multiple factors, including the specific protein, the location and context of the mutation, and the functional importance of the affected region of the protein.

### 3.7. Related Work

In this subsection, we review the existing state-of-the-art research work related to the problem of predicting the pathogenicty of a somatic variant in the myeloid panel. We discuss the strengths and weaknesses of previous methods, focusing on their performance metrics such as accuracy, precision, specificity, and sensitivity (Table 11 and Figure 16).

Several studies have investigated the application of machine learning algorithms to address variants pathogenicity. For instance, we applied random forest to predict the pathogenicty in the solid tumors somatic cancer [1] and to the myeloid panel. According to the Table 11, we can analyze the strengths and weaknesses of each model as follows:Random forest:−Strengths:*High accuracy of 98.50% indicates that the model performs well in classifying the data.*High precision of 97.06% suggests that the positive predictions (class 1) are mostly accurate.*High specificity of 97.01% indicates that the model has a low false positive rate.−Weaknesses:*Although the sensitivity is 1.00 (indicating perfect detection of positive cases), this high sensitivity could also imply a high false negative rate.XGBoost (DE):−Strengths:*Very high accuracy of 99.35% suggests excellent overall performance.*High precision of 98.70% indicates accurate positive predictions.*High specificity of 98.71% suggests a low false positive rate.*Sensitivity of 1.00 indicates perfect detection of positive cases.−Weaknesses:*No specific weaknesses identified.XGBoost (PSO):−Strengths:*High accuracy of 99.04% indicates good overall performance.*High precision of 98.10% suggests accurate positive predictions.*High specificity of 98.11% suggests a low false positive rate.*Sensitivity of 99.98% indicates excellent detection of positive cases.−Weaknesses:*No specific weaknesses identified.XGBoost (GA):−Strengths:*Although the accuracy is lower at 92.51%, it still indicates reasonable overall performance.*High precision of 90.01% suggests accurate positive predictions.*High specificity of 90.02% suggests a low false positive rate.*Sensitivity of 95.03% indicates good detection of positive cases.−Weaknesses:*Lower accuracy and sensitivity compared with other models.XGBoost (SA):−Strengths:*High accuracy of 99.13% indicates excellent overall performance.*High precision of 98.28% suggests accurate positive predictions.*High specificity of 98.28% suggests a low false positive rate.*Sensitivity of 99.98% indicates excellent detection of positive cases.−Weaknesses:*No specific weaknesses identified.

Based on this analysis, we chose XGBoost tuned with DE as the preferred model, because it consistently achieves high performance across all metrics (accuracy, precision, specificity, and sensitivity) without any specific weaknesses identified. It provides the highest accuracy of 99.35% and achieves perfect detection of positive cases (sensitivity of 1.00).

## 4. State-of-the-Art Comparison

In this section, we compare the performance of our proposed approach, which utilizes the Extreme Gradient Boosting (XGBoost) algorithm tuned with metaheuristic algorithms, with the existing state-of-the-art approaches for predicting the pathogenicity of myeloid NGS onco-somatic variants. We focus on studies that have employed different algorithms and metrics to assess the predictive performance.

One notable study by Smith et al. [44] investigated the role of gene mutation profiles in differentiating AML with myelodysplasia-related changes (AML-MRC) from non-MRC AML. They utilized a Next-Generation Sequencing (NGS) panel covering 53 AML-related genes and applied supervised tree-based classification models, including decision tree, random forest, and XGBoost, as well as logistic regression. The results demonstrate that all methods achieved good accuracy in differentiating AML-MRC from non-MRC AML, with the area under the curve (AUC) of the receiver operating characteristics (ROC) ranging from 0.69 to 0.78. Particularly, the XGBoost algorithm showed the best performance, with an AUC of 0.78, outperforming the other methods.

In another study by Johnson et al. [45], the researchers aimed to discover the source of fever in leukemia patients with fever of unknown origin (FUO) using machine learning algorithms. They applied the XGBoost algorithm to a big data repository of leukemia patients and compared its performance with other machine learning algorithms. The results indicate that XGBoost achieved the best performance, with an AUC of 0.8376 and an F1-score of 0.7034, outperforming the other algorithms employed in the study.

These studies highlight the effectiveness of the XGBoost algorithm in differentiating AML-MRC from non-MRC AML and predicting the source of fever in leukemia patients with FUO. In our study, we build upon this existing literature by applying XGBoost and further optimizing its hyperparameters using metaheuristic algorithms. We assess its performance using a comprehensive set of evaluation metrics, including accuracy, precision, sensitivity, specificity, and AUC to provide a robust analysis of its predictive capabilities for myeloid NGS onco-somatic variant pathogenicity.

By comparing our results with the state-of-the-art studies mentioned above, we demonstrate that our optimized XGBoost approach achieves competitive performance and further improves upon the existing methods for predicting the pathogenicity of myeloid NGS onco-somatic variants.

## 5. Discussion

Our goal was to use metaheuristic algorithms to optimize the hyperparameters of the Extreme Gradient Boosting algorithm in the prediction of the pathogenicity of mutation variants in our myeloid panel. We employed four metaheuristic algorithms, namely Differential Evolution (DE), Particle Swarm Optimization (PSO), Genetic Algorithm (GA), and Simulated Annealing (SA), and evaluated their performance based on the accuracy, kappa, and runtime on the training set. The results of the training set showed that DE achieved the highest accuracy (99.22%) and kappa (98.70%) among the four algorithms, with a runtime of 6.64 h. PSO and SA also achieved high accuracy (99.04%) and kappa (98.08%) with much shorter runtimes of 10.21 min and 5.30 h, respectively. GA, on the other hand, achieved a lower accuracy (93.33%) and kappa (84.63%), with a runtime of 2.36 h. We further evaluated the performance of the optimized XGBoost models on the test set using various performance metrics, including accuracy, precision, specificity, sensitivity, Matthew Correlation Coefficient (MCC), area under the curve (AUC), F1-score, and error rate. DE achieved the highest accuracy (99.35%), precision (98.70%), AUC (99.78%), and MCC (98.70%) on the test set, indicating its superior performance in predicting the pathogenicity of mutation variants in our myeloid panel. PSO and SA also achieved high accuracies (99.04% and 99.13%, respectively) and AUCs (99.79% and 99.80%, respectively), while GA had lower accuracy (92.51%), precision (90.01%), AUC (97.90%), and MCC (85.13%).

With the most efficient and accurate architecture returned by DE, we conducted further analysis and experiments to validate the performance of the XGBoost algorithm in classifying variants. The comparison with the biologist’s decision and other predictive algorithms provides valuable insights into the strengths and weaknesses of the algorithm and highlights the importance of domain knowledge in variant classification. Out of 89 mutations, XGBoost agreed 88 times with the biologist’s decision. Regarding the discrepancy in classifying the mutation on exon 9 of the CALR gene, it is important to note that variant interpretation is often a complex and subjective process that involves considering multiple lines of evidence, including functional and clinical data. While bioinformatics tools can provide useful predictions of the potential impact of an amino acid change, the final classification of a variant should be based on a comprehensive evaluation of all available evidence. It is also interesting to note the XGBoost algorithm’s ability to correctly classify the JAK2 hotspot mutation as pathogenic, despite other predictive algorithms classifying it as benign. This highlights the potential of machine learning algorithms to identify patterns and features that may not be immediately apparent to humans and underscores the importance of using multiple approaches and sources of data in variant classification.

Furthermore, the algorithm has shown its ability to predict our solid tumor panel hotspots. This result is promising, and we will conduct a test campaign on the performance of the algorithm to classify mutations outside the myeloid panel. Our additional analysis provides valuable insights into the performance and potential limitations of the XGBoost algorithm in variant classification and highlights the importance of incorporating domain knowledge and multiple sources of data in this process. In a previous study, we used metaheuristic algorithms to optimize the architecture of neural networks (MultiLayer Perceptron) [14], and we know that they can help to find an optimal architecture at a lower cost. Metaheuristic algorithms are optimization techniques that are used to find near-optimal solutions to complex problems, including the tuning of machine learning models. Unlike grid search or random search, which are exhaustive and deterministic, they are more flexible and incorporate elements of stochasticity and heuristics to search the solution space more efficiently. The advantages of metaheuristic algorithms in comparison with grid search or random search include a wider range of optimization problems, including those with constraints, multiple objectives, and noisy data. They are also designed to search the solution space more efficiently than grid search or random search. They often use population-based or swarm-based search strategies that allow for parallel exploration of the solution space. They offer significant advantages over grid search and random search in three main areas: global optimization, handling complex objective functions, and automated stopping criteria. By using population-based or swarm-based search strategies, metaheuristic algorithms are designed to find global optimal solutions, whereas grid search and random search can get stuck in local optima. Additionally, they can handle complex and nondifferentiable objective functions, which can be difficult for grid search and random search. Finally, metaheuristic algorithms often have automated stopping criteria, such as when a satisfactory solution is found or when a maximum number of iterations is reached, which can save time compared with a grid search or random search that require manual stopping.

One advantage of XGBoost is its ability to handle large datasets with high-dimensional features, which is common in bioinformatics applications such as genomic data analysis [46]. XGBoost also provides built-in feature selection capabilities, which can help to identify the most important variables for prediction [47]. However, optimizing hyperparameters for XGBoost can be a time-consuming and computationally intensive process. This is where metaheuristic algorithms such as Differential Evolution can provide significant advantages, as they can help to reduce the search space and find optimal solutions more efficiently [48].

Our study demonstrates that metaheuristic algorithms, particularly DE, PSO, and SA, are effective in optimizing the hyperparameters of the XGBoost algorithm for predicting the pathogenicity of mutation variants in our myeloid panel. The high accuracy, precision, AUC, and MCC achieved by the optimized models on the test set show their potential for clinical application. However, it should be noted that further testing is required to evaluate their performance on mutations outside the myeloid panel. Overall, our results suggest that metaheuristic algorithms are valuable tools for optimizing machine learning models in bioinformatics applications, especially when dealing with high-dimensional and complex datasets.

Moreover, the social implications of our study are noteworthy. The accurate prediction of mutation pathogenicity contributes to a deeper understanding of genetic diseases and their underlying mechanisms. This knowledge can facilitate the development of targeted therapies and interventions, leading to better patient outcomes and improved healthcare strategies. Additionally, by incorporating metaheuristic algorithms in bioinformatics research, we promote the adoption of advanced computational techniques that enhance the field’s efficiency and effectiveness.

One potential avenue for further research is to incorporate domain knowledge or other data modalities into the model. For example, in the context of predicting the pathogenicity of mutations, domain knowledge about the molecular mechanisms of diseases or the functional impact of mutations could be used to guide the feature selection or the hyperparameter optimization process. Additionally, incorporating other data modalities, such as gene expression or protein–protein interaction networks, could potentially improve the accuracy of the model. However, integrating multiple data modalities can be challenging, and it may require developing new algorithms or models that can handle the complexity of the data.

## 6. Limitations

While our study provides valuable insights into the optimization of the XGBoost algorithm for predicting the pathogenicity of mutation variants in our myeloid panel, it is important to acknowledge several limitations.

### 6.1. Dataset Limitations

One limitation of our study is the reliance on a specific dataset comprising mutation variants in the myeloid panel. The generalizability of our findings to other datasets or different types of cancer may be limited. The performance of the optimized XGBoost models should be further validated on diverse datasets to assess their robustness and applicability to broader populations.

### 6.2. Feature Selection

Our study focused on optimizing the hyperparameters of the XGBoost algorithm, assuming that the input features are appropriately selected and preprocessed. However, the performance of the model heavily relies on the quality and relevance of the features used for prediction. The selection of relevant features remains a challenging task, and the inclusion of additional data modalities or domain knowledge could potentially improve the model’s accuracy and generalizability.

### 6.3. Domain-Specific Interpretation

Variant classification is a complex task that involves considering various lines of evidence, including functional and clinical data. While our study focused on the predictive performance of the XGBoost algorithm, the final classification of a variant should incorporate expert domain knowledge and integrate multiple sources of information. It is crucial to note that our models provide predictions based on the input features but may not capture the entire complexity of variant interpretation.

### 6.4. Computational Resources

Our study employed metaheuristic algorithms for hyperparameter optimization, which can be computationally intensive, especially for large datasets. The runtime required for optimizing the XGBoost algorithm may limit its practical implementation in real-time or resource-constrained environments. Efficient parallelization or algorithmic optimizations may be required to overcome these computational limitations.

### 6.5. External Validation

Although we performed rigorous evaluations on our test set, external validation on independent datasets from different populations or clinical settings is necessary to assess the generalizability and reliability of our models. Future studies should aim to validate the performance of the optimized XGBoost models on diverse datasets to ensure their robustness and effectiveness in real-world scenarios.

### 6.6. Error Sequencing and Implementation Variations

One limitation of the XGBoost algorithm is its assumption of error-free sequencing data. In reality, sequencing technologies may introduce errors and artifacts that can impact the accuracy of variant classification. The algorithm’s performance may be influenced by the quality and reliability of the sequencing data. Additionally, the implementation of the algorithm in different laboratories, cities, or countries may introduce variations in protocols, sample characteristics, and other experimental factors. Therefore, it is essential to evaluate the performance of the model in different implementation settings to assess its generalization and adaptability to new conditions.

## 7. Conclusions

We demonstrate the effectiveness of using metaheuristic algorithms to optimize hyperparameters for the XGBoost algorithm in predicting the pathogenicity of mutation variants in our myeloid panel. The high precision and AUC achieved by our model are promising results that show the potential of this approach for improving diagnostic accuracy in bioinformatics. Further research could explore the use of metaheuristic algorithms for optimizing hyperparameters in other machine learning algorithms commonly used in bioinformatics.

Overall, metaheuristic algorithms offer a powerful and flexible tool for tuning machine learning models and finding near-optimal solutions to complex problems.

## Figures and Tables

**Figure 1 bioengineering-10-00753-f001:**
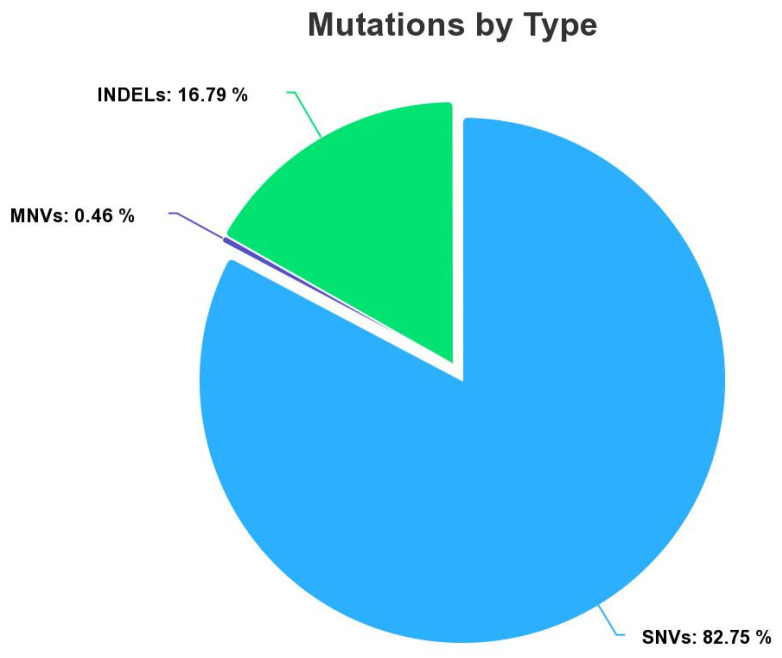
Total SNVs, MNVs, and INDELs in our dataset.

**Figure 2 bioengineering-10-00753-f002:**
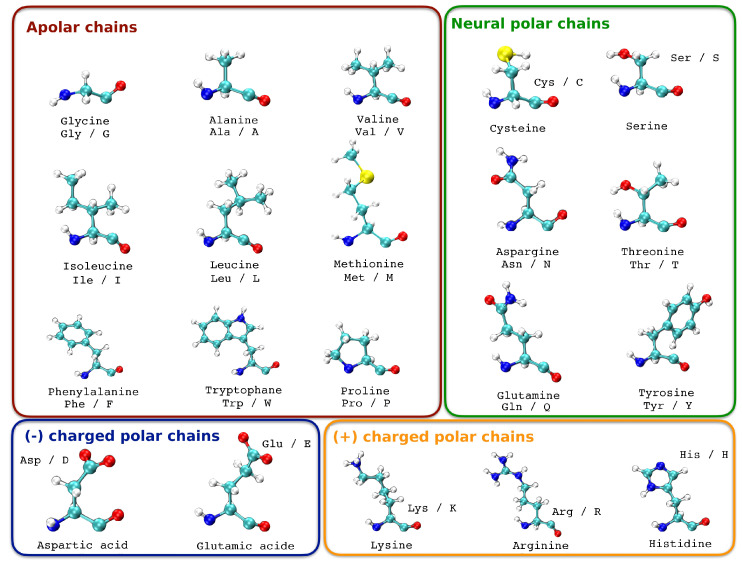
Categorization of amino acids according to the polarity of their side chains.

**Figure 3 bioengineering-10-00753-f003:**
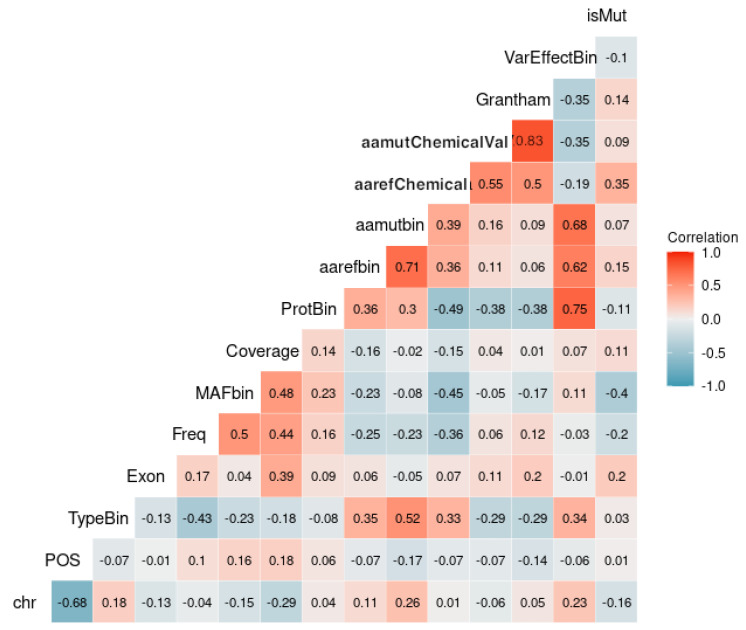
Correlation plot showing the relationship between the variable isMut and other predictor variables in the dataset. The color intensity represents the strength and direction of the correlation, with blue indicating a negative correlation and red indicating a positive correlation. This plot helps to identify which predictor variables are most strongly associated with the outcome variable isMut.

**Figure 4 bioengineering-10-00753-f004:**
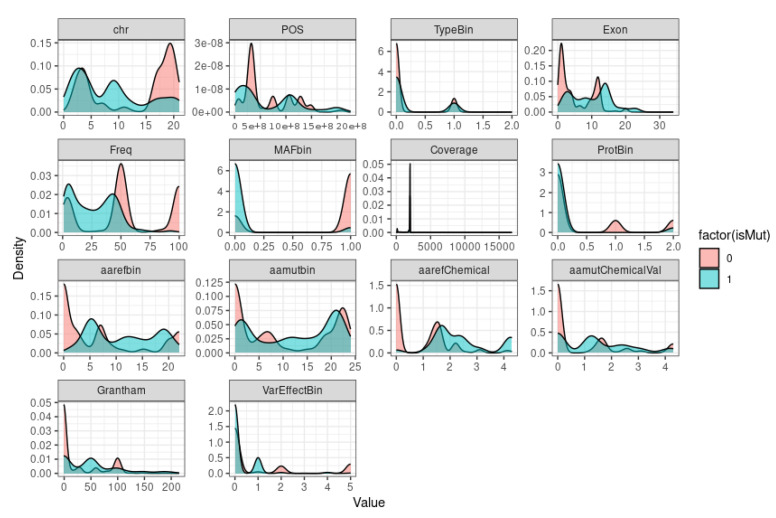
In the density plot, each individual density plot shows the distribution of values for one of the variables in our dataset. The x-axis shows the values that the variable can take, and the y-axis shows the estimated probability density for each value. The plot is split into two colors (blue and red) corresponding to the two possible values for the isMut output column (0 and 1). The darker the color, the higher the density of points in that area.

**Figure 5 bioengineering-10-00753-f005:**
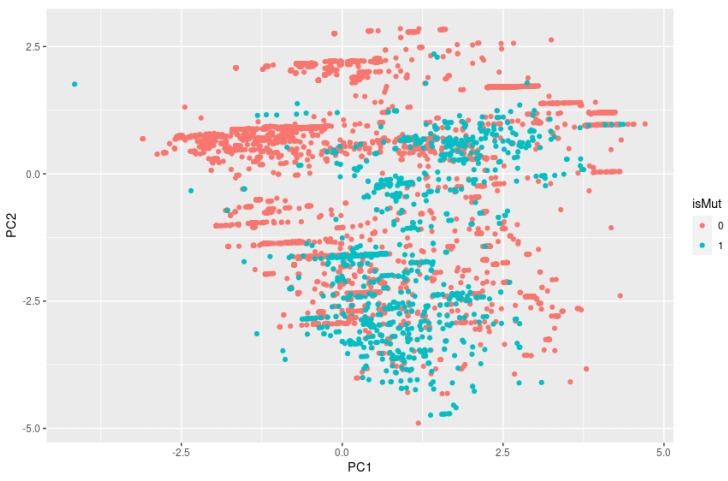
Two-dimensional PCA plot with no clear clustering patterns for the variable isMut. isMut value 0 refers to benign and isMut value 1 refers to pathogenic.

**Figure 6 bioengineering-10-00753-f006:**
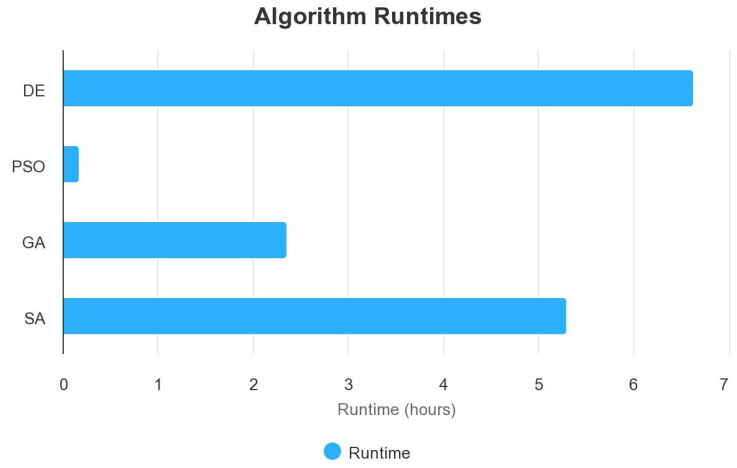
Runtimes algorithm comparison.

**Figure 7 bioengineering-10-00753-f007:**
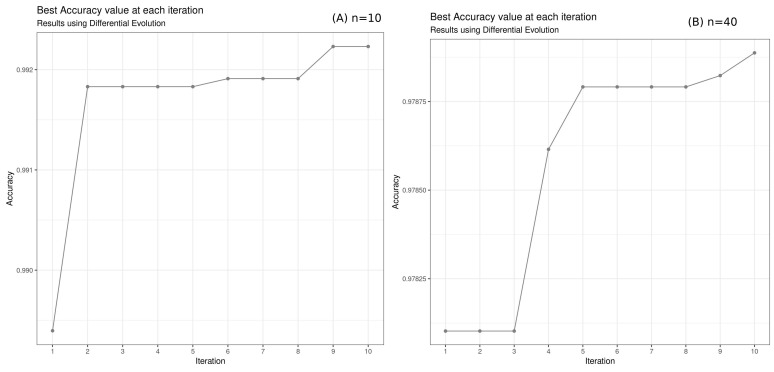
DifferentialEvolution accuracy performance in function of iterations: (**A**) The population size (*n* was fixed to 10. After 10 iterations and 6.64 h of calculation, the maximum accuracy of 99.22% was reached. (**B**) Population size (n=40). After 10 iterations and 1.876445 days of calculation, the maximum accuracy of 0.9790 was reached by the algorithm.

**Figure 8 bioengineering-10-00753-f008:**
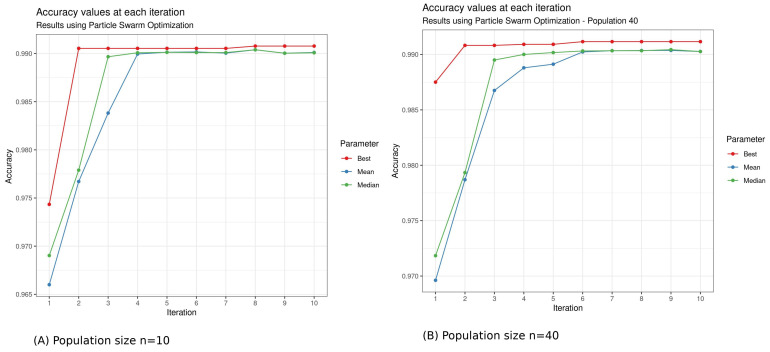
Particle Swarm Optimization accuracy performance in function of iterations: (**A**) The population size (*n* was fixed to 10. After 10 iterations and 10.21 min of calculation, the maximum accuracy of 99.04% was reached by PSO. (**B**) Population size (n=40). After 10 iterations and 43.87407 min minutes of calculation, the maximum accuracy of 0.9912 was reached by the algorithm.

**Figure 9 bioengineering-10-00753-f009:**
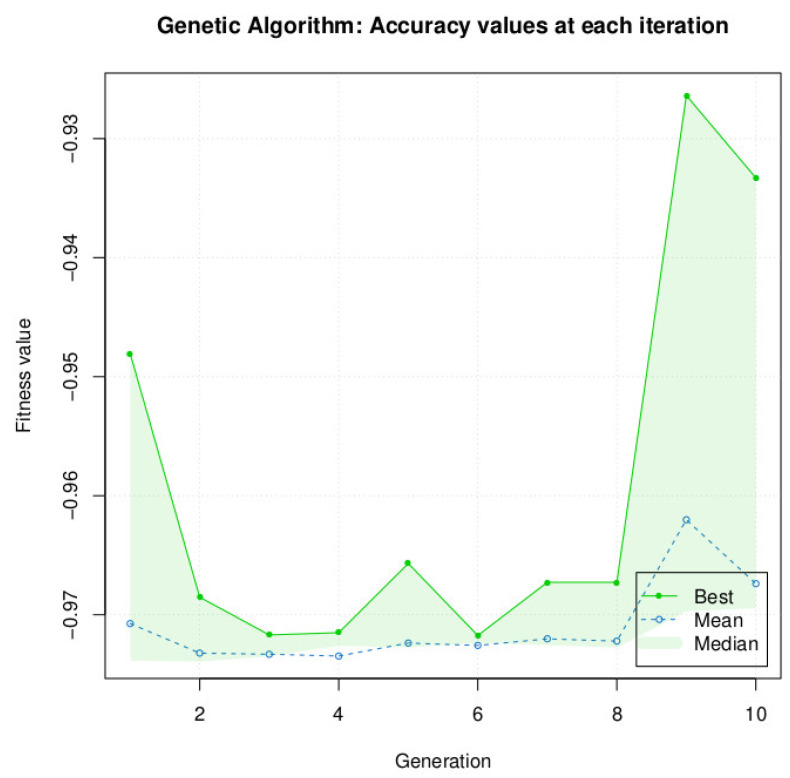
Genetic Algorithm accuracy performance in function of iterations. For the population size, *n* was fixed to 10. After 10 iterations and 5.30 h of calculation, the maximum accuracy of 93.33% was reached by GA.

**Figure 10 bioengineering-10-00753-f010:**
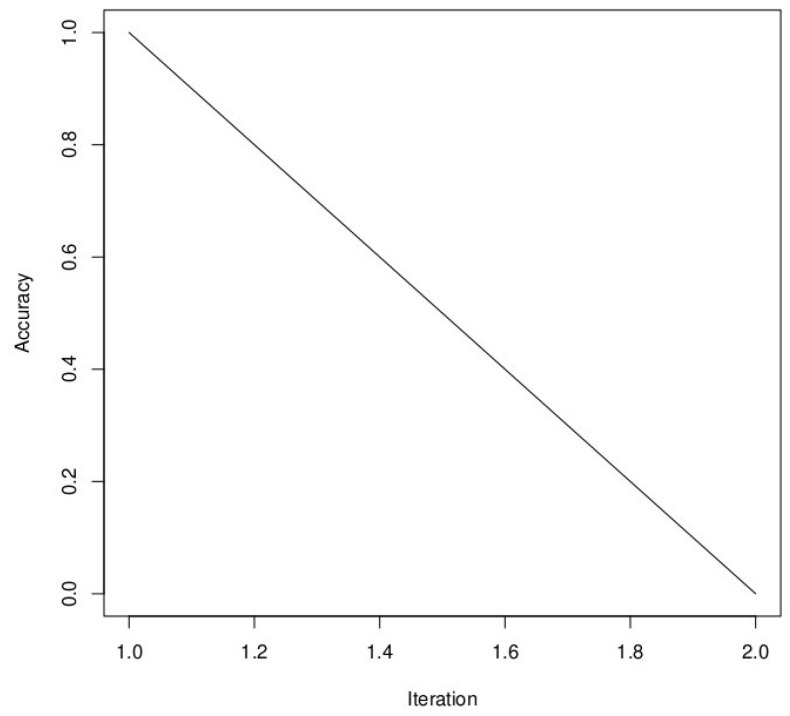
Convergence plot of Simulated Annealing algorithm with maximum iterations = 100. The y-axis represents the value of the objective function (accuracy), while the x-axis range of 0 to 2.0 likely represents the range of temperature decreases over the 100 iterations. The maximum accuracy obtained after 100 iterations was 99.04%.

**Figure 11 bioengineering-10-00753-f011:**
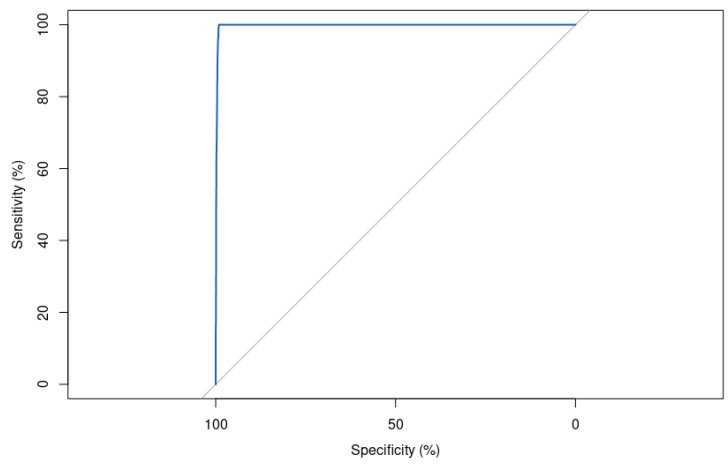
Receiver operating characteristic (ROC) curve for XGBoost model with hyperparameters optimized by Differential Evolution on the test set, showing an area under the curve (AUC) of 99.78. This highly accurate model demonstrates excellent discrimination ability in differentiating between positive and negative samples on the classification task.

**Figure 12 bioengineering-10-00753-f012:**
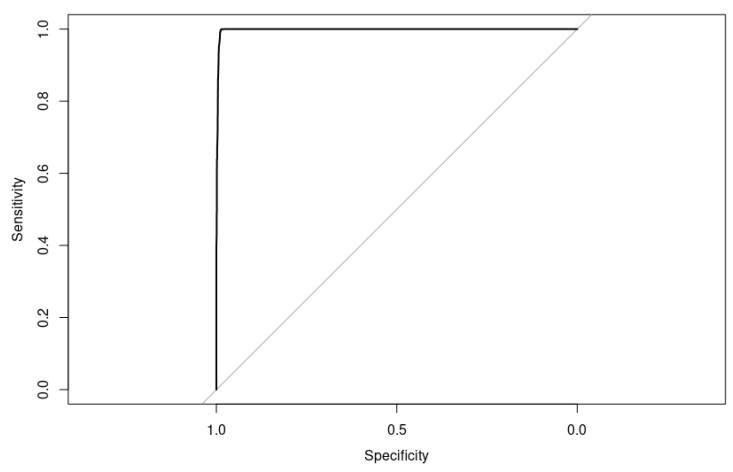
Receiver operating characteristic (ROC) curve for XGBoost model with hyperparameters optimized by Particle Swarm Optimization (PSO) on the test set, showing an area under the curve (AUC) of 99.79. This highly accurate model demonstrates excellent discrimination ability in differentiating between positive and negative samples on the classification task.

**Figure 13 bioengineering-10-00753-f013:**
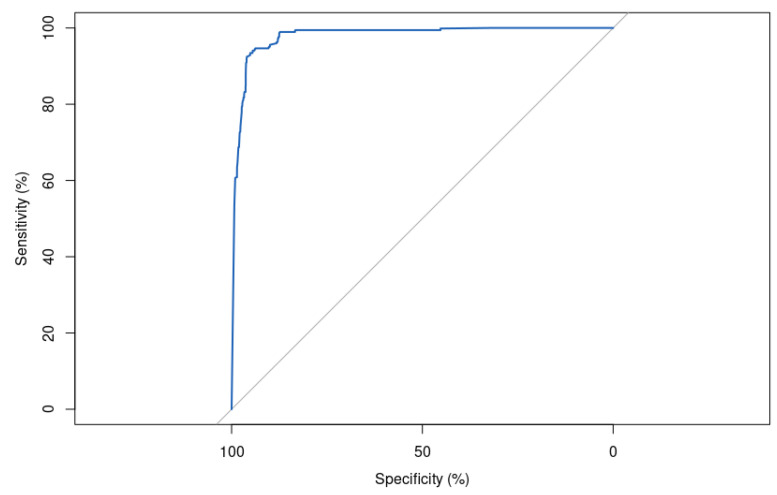
Receiver operating characteristic (ROC) curve for XGBoost model with hyperparameters optimized by Genetic Algorithm (GA) on the test set, showing an area under the curve (AUC) of 97.90%. This highly accurate model demonstrates excellent discrimination ability in differentiating between positive and negative samples on the classification task.

**Figure 14 bioengineering-10-00753-f014:**
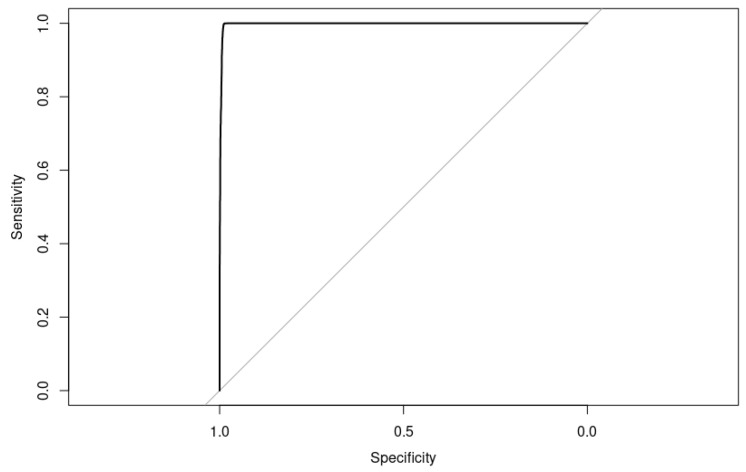
Receiver operating characteristic (ROC) curve for a binary classifier with Simulated Annealing hyperparameters optimization, AUC = 0.998, on the test set.

**Figure 15 bioengineering-10-00753-f015:**
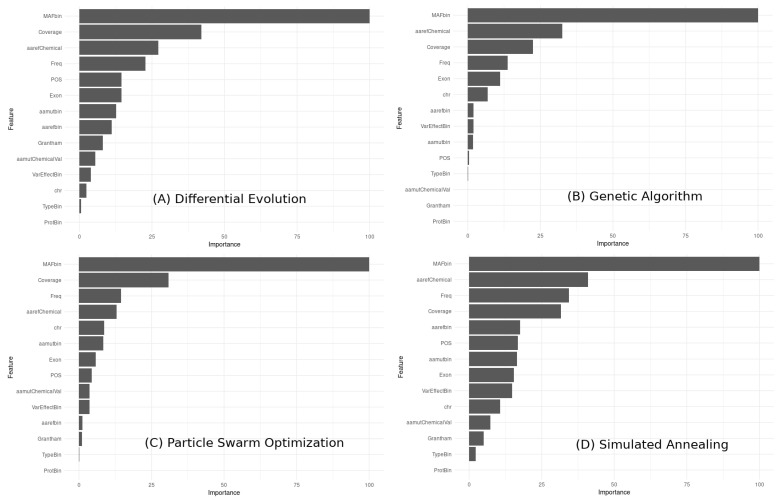
This bar chart shows the feature importance rankings for four metaheuristic algorithms (GA, PSO, DE, and SA) on a binary classification task in medical diagnosis. The importance rankings were computed using the xgbTree variable importance method. The chart indicates that MAFbin was consistently the most important feature across all four algorithms. However, the relative importance of other features varied across the algorithms. Overall, the chart highlights the variability in feature importance rankings obtained from different metaheuristic algorithms.

**Figure 16 bioengineering-10-00753-f016:**
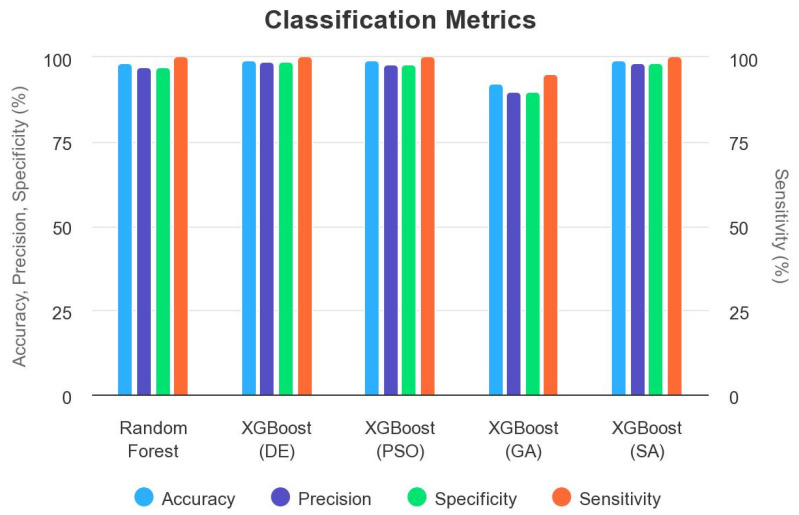
Comparison between random forest vs XGBoost tuned with metaheuristic algorithms.

**Table 1 bioengineering-10-00753-t001:** Feature encoding for training the XGBoost algorithm. Data selected to be passed as a parameter in the Extreme Gradient Boosting algorithm. Some data had to be recoded in order to be recognized by the algorithm. The 14 input parameters are chromosome (chr), POS (position), TypeBin (type of mutation), exon, frequency (freq), Minor Allele Frequency (MAFbin), Coverage, ProtBin, aarefbin (amino acid reference), aamutbin (amino acide after the mutation), aarefCehmical, aamutChemicalVal, Grantham, and varEffectBin. IsMut is the output parameter for the supervised learning (classification task).

Row Names	Data Description	Data Coding	Data Type
chr	Number of the chromosome where the mutation is	No encoding	Integer
POS	Position of in the chromosome where the mutation is	No encoding	Big Integer
TypeBin	Type of mutation: Single Nucletide Variant (SNV), Multiple Nucleotide Variants (MNV), insertion deletion (INDEL)	SNV = 0, MNV = 1, INDEL = 2	Integer
Exon	Exon number. 1 for the first exon in gene, 2 for the second one, …, *n*	exon = {1, …, *n*} exon = 0 when it is intronic. Exon	Integer
Freq	Variant Allele Frequency (frequency of the mutation)	No encoding	Float
MAFbin	Minor Allele Frequency	MAF value if it exists, −1 when there is no MAF	Float
Coverage	Sequencing coverage (0 if <300 reads, 1 if >300 reads)	No encoding	Big integer
Protbin	Mutation effect on the protein (amino acid change)	Intronic or splice site (p.? = 0), same acid amino (p.(=) = 1), acid amino change = 2	Integer
aarefbin	Amino acid (before the mutation)	Arg = 1, His = 2, Lys = 3, Asp = 4, Glu = 5, Ser = 6, Thr = 7, Asn = 8, Gln = 9, Trp = 10, Sec = 11, Gly =12, Pro = 13, Ala = 14, Val = 15, Ile = 16, Leu = 17, Met = 18, Phe = 19, Tyr = 20, Cys = 21	Integer (more detail on Table 2)
aamutbin	Amino acid (after the mutation)	Arg = 1, His = 2, Lys = 3, Asp = 4,Glu = 5, Ser = 6, Thr = 7, Asn = 8, Gln = 9, Trp = 10, Sec = 11, Gly = 12, Pro = 13, Ala = 14, Val = 15, Ile = 16, Leu = 17, Met = 18, Phe = 19, Tyr = 20, Cys = 21, fs = 22, Ter = 22, del =22	Interger (more detail on Table 2)
aarefChemical	Charge of the amino acid side chains (before the mutation)	See Table 3 or Figure 2	Float
aamutChemicalVal	Charge of the amino acid side chains (after the mutation)	See Table 3 or Figure 2	Float
Grantham	Grantham score of the mutation	Grantham value from 5 to 215. −1 when it is not applicable	Grantham
varEffectBin	Effect of the mutation on the reading frame (frameshift, missense …)	Frameshitf = 3, missense = 1, nonsense = 2, synonymous = 0, unknown = −1	Variant.effect
isMut	Potential pathogenic variant. Decision of the biologist on the mutation variant	Benign/uncertain significance = 0, pathogenic = 1	Boolean

**Table 2 bioengineering-10-00753-t002:** Chemical properties of amino acids. This table groups amino acids according to their chemical properties.

Chemical Properties	Amino Acids
Acidic	Aspartic (Asp), Glutamic (Glu)
Aliphatic	Alanine (Ala), Glycine (Glycine), Isoleucine (Ile), Leucine (Leu), Valine (Val)
Amide	Asparagine (Asn), Glutamine (Gln)
Aromatic	Phenylalanine (Phe), Tryptophan (Trp), Tyrosine (Tyr)
Basic	Arginine (Arg), Histidine (His), Lysine (Lys)
Hydroxyl	Serine (Ser), Threonine (Thr)
Imino	Proline (Pro)
Sulfur	Cysteine (Cys), Methionine (Met)

**Table 3 bioengineering-10-00753-t003:** Classification of Amino Acids based on Polarity of Side Chains.

Group	1 Letter Code	3 Letters Code	Encoded Value
Apolar	A	Ala	1.1
Apolar	F	Phe	1.2
Apolar	I	Ile	1.3
Apolar	L	Leu	1.4
Apolar	M	Met	1.5
Apolar	P	Pro	1.6
Apolar	V	Val	1.7
Apolar	W	Trp	1.8
Apolar	G	Gly	1.9
Uncharged	C	Cys	2.1
Uncharged	N	Asn	2.3
Uncharged	Q	Gln	2.4
Uncharged	S	Ser	2.5
Uncharged	T	Thr	2.6
Uncharged	Y	Tyr	2.7
Negative charged	D	Asp	3.1
Negative charged	E	Glu	3.2
Positive charged	H	His	4.1
Positive charged	K	Lys	4.2
Positive charged	R	Arg	4.3
NA	NA	Ter	0.0
NA	NA	dup	0.0
NA	NA	del	0.0
NA	NA	p. ?	0.0
NA	NA	p.(=)	0.0
NA	NA	fs	0.0

**Table 4 bioengineering-10-00753-t004:** Genetic Algorithm parameters. Elitism remains the proportion of the best individuals selected for the next generation. Random selection corresponds to the proportion of random individuals selected for the next generation. This ensures maintaining a significant genetic diversity in each generation and avoids converging on a local minimum. The population size must be significant enough to include the maximum number of solutions. The mutation rate defines the frequency at which genes will mutate. The number of generations will define the limits of the algorithm’s research framework.

Parameters	Value
Elitism	0.3
Population size	10
Random selection	0.1
Mutation rate	0.5
Number of generations	10

**Table 5 bioengineering-10-00753-t005:** Interpretation of Cohen’s kappa values.

Value of Kappa	Level of Agreement	% of Data That Are Reliable
0–0.20	None	0–4
0.21–0.39	Minimal	4–15
0.40–0.59	Weak	15–35
0.80–0.90	Strong	64–81
Above 0.90	Almost perfect	82–100

**Table 6 bioengineering-10-00753-t006:** Performance metrics obtained with the four metaheuristic algorithms on XGBoost and values of hyperparameters found: nrounds (the number of boosting iterations to perform), eta (learning rate or step size for shrinking the contribution of each tree in the model), md (max_depth, the maximum depth of each tree in the model), γ (minimum reduction in the loss function required to split a node in the tree), cols (colsample_bytree, the fraction of columns (features) to be used in each split), mw (min_child_weight, the minimum number of samples required in a leaf node), sub(subsample, the fraction of the training set to use for each boosting iteration), and runtime (time taken by the algorithm to find the hyperparameters).

Method	Accuracy	Kappa	Nrounds	Eta (10⌃x)	Md	γ	Cols	Mw	Sub	Runtime
DE	99.22	98.70	186	−0.36	4.71	0.25	0.73	1.49	0.54	6.64 h
PSO	99.04	98.08	149	−1	6	1	1	1	1	10.21 min
GA	93.33	84.63	120	−2.75	2.49	0.19	0.66	2.04	0.78	2.36 h
SA	99.04	98.08	175	−1.04	5.67	0.08	0.60	2.36	0.56	5.30 h

**Table 7 bioengineering-10-00753-t007:** Summary of the performance metrics of the 4 metaheuristic algorithms on the test set.

Method	Accuracy	Precision	Specificity	Sensitivity	MCC	AUC	F1-Score	Err
DE	99.35	98.70	98.71	1	98.70	99.78	99.35	0.65
PSO	99.04	98.10	98.11	99.98	98.08	99.79	99.03	0.96
GA	92.51	90.01	90.02	95.03	85.13	97.90	92.65	7.49
SA	99.13	98.28	98.28	99.98	98.26	99.80	99.12	0.87

**Table 8 bioengineering-10-00753-t008:** Proposed classification for sequence variants identified by XGBoost tuned with Differential Evolution (DE).

XGBoost Probability Score	Description
Inferior to 0.001	Benign
0.001 to 0.049	Likely benign
0.05 to 0.949	Uncertain
0.95 to 0.99	Likely pathogenic
Superior to 0.99	Pathogenic

**Table 9 bioengineering-10-00753-t009:** Comparison of Predictive Algorithms for Genetic Variants: Performance of XGBoost Tuned with Differential Evolution (DE) versus Biologist Decision and Others. The table shows the performance of several predictive algorithms for genetic variants, including XGBoost tuned with Differential Evolution (DE), UMD Predictor, Varsome, and CGI. The genetic variants analyzed include DNMT3A, JAK2, TET2, CEBPA, GATA2, ASXL1, and CALR. The table presents various features of these variants, such as MAF (Minor Allele Frequency), amino acid changes, and coding mutations, as well as whether the variant is pathogenic, benign, or uncertain. The performance of each algorithm is compared against a biologist’s decision, and the table also indicates whether the variant is a driver or a passenger mutation.

Genes	MAF	Amino Acid	Coding	isMut	Xgboost DE	UMD Predictor	Varsome	CGI
DNMT3A	0.224	p.Leu422=	c.1266G>A	0	Benign	Polymorphism	Benign	Passenger
DNMT3A		p.Arg366His	c.1097G>A	1	Likely pathogenic	Pathogenic	Uncertain	Driver
JAK2		p.Val617Phe	c.1849G>T	1	Pathogenic	Polymorphism	Pathogenic	Passenger
TET2		p.Ser1518Ter	c.4553C>A	1	Pathogenic	Pathogenic	Uncertain	Driver
CEBPA		p.Gly104del	c.295_297del	0	Benign	NA	Uncertain	Passenger
GATA2	0.233	p.Ala164Thr	c.490G>A	0	Benign	Polymorphism	Benign	Passenger
CEBPA		p.Ala259Thr	c.775G>A	0	Likely benign	Probable polymorphism	Benign	Passenger
ASXL1	0.02	p.Val751Ile	c.2251G>A	0	Likely benign	Polymorphism	Benign	Passenger
CALR		p.Lys385Thr	c.1154A>C	0	Pathogenic	Probably pathogenic	Benign	Passenger
CALR		p.Lys385IlefsTer	c.1154delinsTTTGTC	1	Pathogenic	NA	NA	Driver

**Table 10 bioengineering-10-00753-t010:** Performance of XGBoost and MLRF models on predicting the pathogenicity of hotspot mutations in solid tumor genes. The table shows the amino acid change, coding mutation, and classification of each mutation as pathogenic or likely pathogenic by XGBoost and MLRF. XGBoost refers to the model tuned with Differential Evolution (DE), while MLRF is the machine learning random forest implemented at the lab. The genes included in the analysis are known hotspots in solid tumors.

Genes	Amino Acid	Coding	XGBoost DE	MLRF
KRAS	p.Gly12Cys	c.34G>T	Pathogenic	NO
KRAS	p.Gln61Lys	c.180_181delinsCA	Pathogenic	YES
KRAS	p.Ala146Pro	c.436G>C	Pathogenic	YES
BRAF	p.Val600Glu	c.1799T>A	Likely pathogenic	YES
BRAF	p.Asp594Asn	c.1780G>A	Likely pathogenic	YES
BRAF	p.Gly469Val	c.1406G>T	Likely pathogenic	YES
EGFR	p.Glu709Lys	c.2125G>A	Likely pathogenic	YES
EGFR	p.Thr790Met	c.2369C>T	Likely benign	YES
EGFR	p.Leu747_Ala750delinsPro	c.2239_2248delinsC	Pathogenic	YES
PIK3CA	p.His1047Arg	c.3140A>G	Pathogenic	YES
EGFR	p.Thr725Met	c.2174C>T	Likely benign	YES

**Table 11 bioengineering-10-00753-t011:** Summary of performance metrics for random forest and XGBoost with metaheuristic algorithms on the myeloid panel.

Method	Accuracy	Precision	Specificity	Sensitivity
Random forest	98.50	97.06	97.01	1
XGBoost (DE)	99.35	98.70	98.71	1.00
XGBoost (PSO)	99.04	98.10	98.11	99.98
XGBoost (GA)	92.51	90.01	90.02	95.03
XGBoost (SA)	99.13	98.28	98.28	99.98

## Data Availability

Training database and R source code are available on demand.

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
