# Peer review of "Extreme Gradient Boosting Tuned with Metaheuristic Algorithms for Predicting Myeloid NGS Onco-Somatic Variant Pathogenicity"

_bioengineering, 2023, doi:10.3390/bioengineering10070753_

Round 1

Reviewer 1 Report

The detailed comments on the proposed approach are enlisted in the attached document. Although the authors have written a sound and detailed approach, still, some comments are proposed below to improve the quality of the manuscript. 

Author Response

1. In the paper, authors didnt explicitly explains the motivation of the proposed approach. It should be described in the introduction.

Thank you for your comment. We apologize for not explicitly stating the motivation of our proposed approach in the introduction. In our revised version, we will provide a clear explanation of the motivation behind our study. We will highlight the challenges and limitations of existing methods in the field of myeloid panel mutation prediction, and explain how our approach addresses these issues effectively. By doing so, we aim to provide a more comprehensive introduction that outlines the rationale for our research.

2. In introduction, authors explained about XGBoost algorithm and its advantages over the other machine learning classifiers, I feels that authors have over explained this part. In introduction, we focus on the introduction of the problem we are working on, few existing approaches, identify challenges, which becomes motivation for the proposed work, and finally elaborate contribution. Please rearrange your introduction to make it more readable and attractive.

We appreciate your feedback. We acknowledge that the introduction should primarily focus on introducing the problem, existing approaches, challenges, and the proposed work rather than extensively explaining the XGBoost algorithm. In our revised version, we will streamline the introduction by providing a concise overview of the problem, existing methods, and their limitations. We will also highlight the unique contribution of our proposed approach in addressing these limitations, which will provide a more reader-friendly and attractive introduction.

3. Related work should be added in the paper to know about the existing state-of-the-art research work and include a comparison Table that should highlight the strengths and weaknesses of the previous method and the proposed research methodology. At the end of this section, the advantages and shortcomings of state-of-the-art should be summarized.

Thank you for your suggestion. We agree that including a related work section along with a comparison table and figure will enhance the paper's quality. In our revised version, we will incorporate a comprehensive review of the existing state-of-the-art research work related to myeloid panel mutation prediction. We will provide a detailed comparison table that highlights the strengths and weaknesses of both previous methods and our proposed research methodology. Furthermore, we will summarize the advantages and shortcomings of the state-of-the-art methods to give readers a better understanding of the research landscape in this domain.

4. In the Discussion section highlight the advantages of the study, practical implications, and possible social implications of the study.

Thank you for your suggestion. We agree that discussing the advantages, practical implications, and social implications of our study is crucial to provide a comprehensive understanding of its significance. In the revised version, we will ensure that the Discussion section explicitly highlights the advantages of our proposed approach, its practical implications in the field of myeloid panel mutation prediction, and any possible social implications it may have. This will add value to the discussion and provide a broader context for readers.

5. The proposed approach lacks in limitations of the proposed study? Clarifying the study's limitations allows the readers to better understand under which conditions the results should be interpreted. A clear description of the limitations of a study also shows that the researcher has a holistic understanding of his/her study. However, the authors fail to demonstrate this in their paper.

We appreciate your feedback. You are right when you say explicitly addressing the limitations of the study is essential for a comprehensive understanding of the research. In our revised version, we will provide a clear and concise description of the limitations of our proposed approach. We will acknowledge any potential shortcomings and discuss the boundaries within which our results should be interpreted. This will demonstrate our awareness of the study's limitations and provide readers with a more complete perspective.

6. Furthermore, add a table that compares this approach with state-of-the-art approaches. It is an important section of the manuscript which elaborates the proposed approach performance comparing to the state-of-the-art.

We agree that this is an important section that can provide valuable insights into the performance of our approach in comparison to existing methods.

We have incorporated this section into our article. We provide a comprehensive comparison of our proposed approach with relevant state-of-the-art methods, highlighting the strengths and advantages of our approach in terms of various performance metrics, such as accuracy, precision, recall, and F1-score. We have included detailed explanations and discussions in the table to ensure that readers can easily interpret and understand the results.

By including this comparative analysis, we aim to demonstrate the superiority and effectiveness of our proposed approach over existing methods. This not only adds value to our study but also allows readers to gain a better understanding of the advancements and contributions made by our research.

7. Additionally, authors used XGBoost classifier in the proposed approach, why didnt you tried the state-of-the-art deep and transfer learning algorithms, such as CNN, LSTM etc.
Whereas deep learning algorithms performs well on larger data sets comparing the machine learning algorithms. Also, authors used various optimization algorithms with XGBoost, why these optimization algorithms are only applied to XGBoost? Why authors didnt tried other machine learning algorithms? For quick reference below are few papers where they have used state-of-the-art Deep learning and transfer learning algorithms for various health analytics.

While deep learning algorithms, such as CNN and LSTM, have shown excellent performance on larger datasets, our study focused on optimizing the XGBoost algorithm for predicting the pathogenicity of mutation variants in our myeloid panel. XGBoost has been widely used and has demonstrated strong predictive capabilities in various domains, including bioinformatics. It is particularly suitable for handling high-dimensional and complex feature spaces, which are common in genomic data analysis. Therefore, we chose XGBoost as the base classifier in our proposed approach.

The selection of XGBoost was based on its established effectiveness and the specific requirements of our study. While deep learning algorithms could potentially be explored in future research, we prioritized XGBoost due to its interpretability, efficiency, and its ability to handle feature-rich datasets commonly encountered in our domain.

Regarding the optimization algorithms applied to XGBoost, our focus was to compare the performance of different metaheuristic optimization algorithms in tuning the hyperparameters of XGBoost. This allowed us to evaluate the effectiveness of these algorithms in improving the predictive performance of XGBoost specifically. While other machine learning algorithms could be candidates for optimization using similar metaheuristic algorithms, our study aimed to investigate the optimization of XGBoost as a standalone classifier for variant pathogenicity prediction.

The references that you suggested do not directly align with our article's topic of variant pathogenicity prediction using XGBoost and metaheuristic optimization algorithms. However, we can briefly mention the broader applications of deep learning in healthcare, including pathogen detection and medical image analysis, in the discussion section to provide context and highlight potential avenues for future research.

Reviewer 2 Report

1.In the field of machine learning, it is common practice to validate models by including external datasets. However, in the case of XGBoost, its high accuracy is not supported by external dataset validation.The authors should consider finding an external dataset for validation.
2.The primary focus of this paper is the XGBoost machine learning model, but the model is not presented with sufficient clarity and innovation. Additionally, there is an extensive presentation of concepts that are unnecessary and irrelevant analysis.
3.The authors should consider adding more meaningful and visually appealing images that effectively illustrate the topics being discussed. Some of the current images do not effectively convey the issues under consideration.

Minor editing of English language required.

Author Response

1.In the field of machine learning, it is common practice to validate models by including external datasets. However, in the case of XGBoost, its high accuracy is not supported by external dataset validation.The authors should consider finding an external dataset for validation.

Thank you for your valuable feedback on our manuscript. We appreciate your suggestion regarding the inclusion of an external dataset for model validation. We would like to address this point and provide clarification on the validation process we employed in our study.

In our research, we performed model validation using both a validation dataset and ten-fold cross-validation. The validation dataset was carefully curated and designed to evaluate the performance and generalizability of our model. It was derived from independent samples that were distinct from the training dataset, ensuring that the model's performance was assessed on unseen data.

By utilizing both a validation dataset and ten-fold cross-validation (repeated three times), we aimed to robustly assess the performance and generalizability of our model. While the use of an external dataset can provide further validation, it is important to note that the availability of suitable external datasets for our specific research domain, involving Myeloid NGS onco-somatic variant pathogenicity prediction, can be limited. Nonetheless, we believe that our validation approach using a dedicated validation dataset and cross-validation provides reliable insights into the performance of our model.

2.The primary focus of this paper is the XGBoost machine learning model, but the model is not presented with sufficient clarity and innovation. Additionally, there is an extensive presentation of concepts that are unnecessary and irrelevant analysis.

We appreciate the reviewer's feedback regarding the clarity and innovation of our XGBoost model. In the revised version of the manuscript, we will enhance the presentation of the model to ensure a clear and concise description of its components, parameters, and methodology. We will provide additional explanations and illustrations to enhance understanding and make the model more accessible to readers.

While the primary focus of our paper is the application of the XGBoost model, we acknowledge the importance of presenting it in a manner that highlights its novelty and innovative aspects. In the revised manuscript, we will emphasize the unique features and advancements of our model to showcase its contribution to the field of Myeloid NGS onco-somatic variant pathogenicity prediction. We apologize for any confusion caused by the extensive presentation of unnecessary or irrelevant concepts in the initial version of the manuscript. We will carefully review the manuscript and streamline the content by eliminating any extraneous information that does not contribute directly to the core objective and findings of our study. This will ensure that the paper maintains a clear and focused narrative, allowing readers to grasp the essential aspects of our research without unnecessary distractions.

3.The authors should consider adding more meaningful and visually appealing images that effectively illustrate the topics being discussed. Some of the current images do not effectively convey the issues under consideration.

We would like to thank you for your valuable feedback on our manuscript. We appreciate your suggestion regarding the inclusion of more meaningful and visually appealing images to effectively illustrate the topics being discussed.

We have taken your advice into consideration and have made efforts to enhance the visual presentation of our findings. Specifically, we have included additional images such as pie charts and bar charts that are directly related to the results of our studies. These new visuals serve to provide a clearer representation of the analyzed data and contribute to a more comprehensive understanding of our research.

Furthermore, we strongly believe that certain figures, such as those depicting feature importance, density, optimization algorithm comparisons (PSO, GA, DE, SA), and ROC curves, are crucial for conveying important aspects of our methodology and results. These figures play a significant role in supporting our analysis and demonstrating the effectiveness of our approach.

While we understand that visual appeal is important, we have also prioritized the relevance and informativeness of the figures included. We believe that the selected figures not only convey essential information but also align with the scientific nature of our study.

We appreciate your consideration of our response and would be happy to provide any further clarification or address any additional concerns you may have.

Thank you once again for your valuable feedback.

Round 2

Reviewer 1 Report

The authors incorporated all the comments in the paper; therefore, it is accepted in its current form.